

# Effects of Stratified Active Layers on the High-Altitude Permafrost Warming: A Case Study on the Qinghai-Tibet Plateau

X. Pan[1], Y. Li[1], Q. Yu[2], X. Shi[3], D. Yang[4], K. Roth[5]

[1]Global Institute for Water Security, University of Saskatchewan, 11 Innovation Boulevard, Saskatoon, SK S7N 3H5, Canada
[2]Laboratory of Frozen Soils Engineering, Cold and Arid Regions Environmental and Engineering Research Institute, Chinese Academy of Sciences, Donggang West Road 320, Lanzhou, 730000, China
[3]CSIRO Land and Water, Christian Laboratory, Clunies Ross Street, Black Mountain, Canberra, Australian Capital Territory, 2601, Australia
[4]National Hydrology Research Centre, Environment Canada, 11 Innovation Boulevard, Saskatoon, SK S7N 3H5, Canada
[5]Institute of Environmental Physics, Heidelberg University, Im Neuenheimer Feld 229, Heidelberg, 69120, Germany

*Correspondence to*: Y. Li (yanping.li@usask.ca)

**Abstract.** Seasonally variable thermal conductivity in active layers is one important factor that controls the thermal state of permafrost. The common assumption is that this conductivity is considerably smaller in the thawed than in the frozen state, $\lambda_t/\lambda_f < 1$. Using a 9-year dataset from the Qinghai-Tibet Plateau (QTP) in conjunction with the GEOtop model, we demonstrate that the ratio $\lambda_t/\lambda_f$ may approach or even exceed 1. This can happen in thick active layers, with thicknesses larger than some 1.5 m, with strong seasonal liquid water content changes. It is additionally furthered by typical soil architectures that may lead to a dry inter-layer. These findings suggest that, given the increase in air temperature and precipitation, soil hydraulic properties, particularly soil architecture in those thick active layers must be taken into account properly in permafrost models.

## 1 Introduction

Along with climate warming, permafrost warming has been widely observed in the Arctic and sub-Arctic as well as in mid-latitude high mountains like the Alps and the Tibetan Plateau (Romanovsky et al., 2013; Harris et al., 2009; Cheng and Wu, 2007). The mean annual ground temperature (MAGT) at a depth of zero annual amplitude is often used to indicate permafrost warming (e.g., Wu et al., 2012). The warming rate is controlled by a variety of factors such as weather regimes, geography/geology and ecosystems. Generally, the cold permafrost has a higher warming rate than the warm permafrost (Wu et al., 2012). However, permafrost temperature can differ greatly in the same region due to local factors like topography, soil properties and vegetation. Their responses to climate change is thus also expected to differ. For instance, the permafrost along the Qinghai-Xizang (Tibet) Railway experienced a mean warming rate of 0.02˚C yr⁻¹ at a depth of 6.0 m over the period from 2006 to 2010, and the highest warming rate even reached 0.08˚C yr⁻¹ in the Fenghuo Mts. area (Wu et al., 2012). Given the same change in climate variables, these local factors still cause the underlying permafrost to develop differently.



For instance, one recent study in the central QTP shows that permafrost at 10 sites experienced highly differing warming rates over the period of 2002-2014 (Wu et al., 2015). Even though there was no extraordinary increase in air temperature (0.02˚C yr$^{-1}$), the permafrost temperature at 10-m depth increased at an average rate of 0.01˚C yr$^{-1}$. Wu et al. (2015) suggested that this was due to the increasing rainfall and the asymmetrical seasonal changes in subsurface soil temperatures.

Reflecting the high warming rate of the permafrost on the warming of the atmosphere, some 0.02˚C yr$^{-1}$, we expect a dominating role of subsurface processes in the active layer that amplify the climate warming input.

As a buffer layer, the active layer regulates the energy transfer between atmosphere and permafrost in addition to vegetation and snow cover. In this study, we focus on the high-altitude permafrost on the QTP, which is characterized by a thick unsaturated active layer and sparsely vegetated surface. In contrast to the commonly thin active layers in the Arctic, the

active layers on the QTP are usually over 1.5 m thick. A further important difference between the permafrost in the Arctic and on the QTP is that the latter has a strong diurnal forcing with some 180 freeze-thaw days (Yang et al., 2007). Furthermore, with precipitation concentrated in the summer season as rainfall, the subsurface soil hydraulic properties play a key role in ground heat transfer, in addition to the soil thermal properties. In contrast to the well-studied permafrost in the Arctic, the applicability of the analytic models of the climate-permafrost relationship basing on the seasonally variable

thermal conductivity might be challenging. For instance, TTOP (mean annual temperature at the top of the permafrost table) and related concept of thermal offset, namely the TTOP minus the MAGST (mean annual ground surface temperature) (Smith and Riseborough, 1996; Smith and Riseborough, 2002) are strongly influenced by the seasonally variable thermal conductivity. Normally, a wet active layer has a larger thermal offset than a dry active layer, which has small or even vanishing thermal offset (e.g., Romanovsky and Osterkamp, 1995; Hasler et al., 2011). However, a reversed thermal offset,

namely TTOP > MAGST, has been reported by Lin et al. (2015) on the QTP. To evaluate the applicability of this concept, further exploration of the hydraulic and thermal mechanisms in the active layers on the QTP is highly demanded. This might facilitate us to understand the impact of the active layer on the permafrost warming.

In this study, we use observations over a nine-year period and numerical simulations to investigate a recent permafrost warming at a site in a warm permafrost region on the QTP and to demonstrate the role of a typical stratified active layer in

permafrost warming. Our goals are (1) to diagnose the quick permafrost warming at the study site, (2) to reveal the unique phenomenon of the reversed seasonally variable thermal conductivity in the active layer that challenges the application of the analytic models by comparing with observations and physically based modeling, and (3) to emphasize the importance of incorporating structural soil hydraulic properties in permafrost projections given a rain-dominated weather pattern on the QTP.



## 2 Material and methods

### 2.1 Site descriptions

The Chumaer site is located on a high plain of the Chumaer River catchment in the northeastern QTP with an average altitude of over 4450 m (Fig. 1). In the catchment area, the land surface is covered by bare soil or sparse vegetation. Measurements at the study site comprise a monitoring station and several boreholes, with discontinuous ground temperature measurements since 2006 (Pan et al., 2014). The monitoring station has been complemented by soil-weather measurements since 2006. The weather data from 2006 to 2014 show an average air temperature of about -16.0˚C in January and 7.2˚C in July, and precipitation is dominated by summer monsoon from June to September, which brings about 350 mm precipitation annually, falling mostly as rainfall. Irregular thin snow cover occurs in late spring or early winter, lasting usually just a few weeks. The stratigraphy includes a fine top soil (30 cm) and a middle layer of alluvial sandy and gravelly sediment up to 3 m that lies over deeply weathered mudstone. The borehole data indicate that the permafrost has a thickness about 25 m, and the temperature at a depth of 10 m is less than 1.0˚C. The active-layer thickness is around 2.5 m.

### 2.2 Surface-subsurface monitoring scheme Subsection

The surface-subsurface interaction has been investigated since 2006. Regular meteorological variables including air temperature, precipitation, relative humility, wind speed and direction and net radiation were monitored at an automatic weather station. Subsurface hydraulic and thermal dynamics within the active layer were monitored by measuring soil temperature and soil water content at a variety of depths (soil temperature: 0.05, 0.10, 0.15, 0.20, 0.30, 0.50, 0.70, 0.90, 1.10, 1.30, 1.50, 1.70, 1.92, 2.08, 2.18, 2.30, 2.50, 2.70, 3.00, 3.30, 3.60 m; Soil water content: 0.10, 0.20, 0.40, 0.65, 0.89, 1.19, 1.54, 1.92, 2.10 m). They were recorded with a time interval of 60 minutes. Soil temperature was measured with thermistors, which provides an accuracy of 0.05˚C. Liquid soil water content was measured with CS616 sensors (Campbell Scientific Ltd.), and the total water content in frozen soils was deduced from the value measured just before freezing. Here it assumes negligible soil water redistribution during freezing due to the coarse soils. Thereby, its accuracy is around ±5% (Pan, 2011). More detailed technical description of the instrumentation can be found in Pan (2011). In addition, permafrost temperature was investigated with two boreholes. They were about 30 m away in a flat area. The shallow borehole is 15 m in depth and the deep one is 60 m that penetrates through the permafrost.

### 2.3 GEOtop model

We use the GEOtop (version: 1.45) to simulate fluxes of moisture and energy between the atmosphere, surface and soil, and the soil freezing and thawing. GEOtop is a process-based energy- and mass-balance model (Rigon et al, 2006; Endrizzi, 2009), which has a number of advantages for a wide range of permafrost applications. It allows to simulate hydrological fluxes from the energy balance in the complex terrain with snow-covered and snow-free regimes (e.g., Simoni et al., 2008; Endrizzi and Marsh, 2010). For the subsurface, soil temperature and moisture dynamics are simulated using a robust and



energy-conserving model of freezing in variably-saturated soil (Dall'Amico et al., 2011a). It invokes a relation between the soil freezing characteristic and the soil water characteristic and assumes a rigid. The model's versatility allows to investigate the responses of permafrost degradation on the QTP, where permafrost is characterized as a thick and stratified active layer with pronounced hydraulic dynamics due to the rainfall-evaporation dominated land surface fluxes.

Various applications of GEOtop can be found in the literature (Dall'Amico et al., 2011b and Endrizzi et al., 2014). In this study, we apply the model for the single site as a one-dimensional (1-D) simulation, where the spatial factors, e.g., topography and snow, are not important. However, the detailed representation of the subsurface is essential for our research questions. They are introduced in the following subsections.

## 2.4 Model set-up

Considering the features of land surface energy exchange and subsurface hydraulic and thermal dynamics, a 1-D conceptualized model for the study site is setup with GEOtop. Some assumptions for this model are given as: (1) no lateral flows exist like surface runoff and subsurface groundwater flow; (2) surface features like vegetation and soils, and associated parameterization are constant in the long term simulations.

    The simulated stratigraphic profile constitutes three layers according to field drilling. They are sandy loam (0-0.3 m), sand
(0.3 - 3.0 m) and gradually weathered bedrock (3.0 - 30 m). The profile domain was generated with a high resolution in size of 10 cm for the shallow layer (0-3.0 m) and was gradually reduced to 0.5 m and 1.0 m for the lower layer. There are 63 layers in total. In order solely to diagnose the effect of stratified active layer on permafrost degradation, model simulations were driven by the same atmospheric forcing.

### 2.4.1 Input parameters

The required climate forcing for the GEOtop models includes precipitation (snow and rain), air temperature, wind speed, relative humidity, and incoming short and longwave radiation. The bottom boundary conditions for energy and water balance are set as follows. Considering the weak impact of the bottom mudstone on surface water flux, the bottom drainage rate through the mudstone was simply set as zero. Whereas, the bottom thermal condition is essential to the permafrost warming rate, as well as surface energy fluxes. The geothermal flux at the depth of 30 m was determined from the measured ground
temperature gradient ($0.07°C \ m^{-1}$) and calculated thermal conductivity of the mudstone. Given a thermal conductivity of soil matrix $2.0 \ W \ m^{-1} \ K^{-1}$, and a porosity of 0.2 for the mudstone, the geothermal flux at the bottom boundary was set as $0.14 \ W \ m^{-2}$. This high geothermal flux is consistent with other observed values in the same Kunlun Mountains area (Wu et al., 2010).

    Taking the analysis of the sensitivities and uncertainties of the GEOtop by Gubler et al. (2013) into account, we set the
following surface and subsurface parameters: The vegetation coverage for the sparsely-vegetated ground surface is set to 0.3 and the surface roughness length, which is required for the calculation of the turbulent fluxes, to 120 mm. The latter was chosen in agreement with studies on the QTP (Ishikawa et al., 1999; Yang et al., 2008; Ma et al., 2009). The surface





parameters are assumed to be constant during the simulations. The subsurface hydraulic and thermal parameters for the actual soil profile are listed in Table 1. The required van Genuchten parameters were obtained from soil texture information using the neural network routine (Schaap and Bouten, 1996). Soil textures for the two sub-layers (I and II) are available from König (2008). No data are available for the bottom layer. Thus, hydraulic properties for the third layer are assumed to be the same as the typical clay (Domenico and Schwartz, 1990). Both soil thermal conductivity and heat capacity are functions of the four volumetric components: water, ice, air and soil matrix (Dall'Amico et al., 2011a). The bulk thermal conductivity ($\lambda_b$) was estimated with the following equation proposed by Cosenza at al. (2003)

$$\lambda_b = \left[ (1-\Phi)\sqrt{\lambda_m} + \theta_w\sqrt{\lambda_w} + \theta_i\sqrt{\lambda_i} + \theta_a\sqrt{\lambda_a} \right]^2, \tag{1}$$

where, $\lambda_m$, $\lambda_w$, $\lambda_i$ and $\lambda_a$ are thermal conductivities of soil matrix, water, ice and air, respectively; $\Phi$ is soil porosity; $\theta_w$, $\theta_i$, and $\theta_a$ are the volume fractions of water, ice and air, respectively. Thermal properties of the soil matrix were set as common values for different soil types (Farouki, 1986).

### 2.4.2 Simulation protocol

The investigated stratified active layer is commonly found on the QTP along with a "dry inter-layer" between top soil and bottom soil. While the soil water distribution is mainly related to the unique soil architecture (A1) that a fine-grained layer without (or only thin) surface organic horizons overlaying the coarse immature soils, which is characterized as low content of fine-grained materials like silt and clay (Huang et al., 2006). The role of soil architecture in the stratified active layer in regulating hydraulic and thermal dynamics in active layer, as well as long term permafrost change is investigated with following numerical simulations. For comparison, another two reference active layer with soil architectures A2 and A3 are also used. The references A2 and A3 consist of a single layer for the shallow soils (0-3.0 m) but with fine (I) and coarse (II) minerals, respectively. Since the focus of this study is to investigate the effect of soil architecture on permafrost warming, the thermal conductivity of soil matrix is set as the same value when comparing different soil architectures. Apart from the actual case of soil matrix with high thermal conductivity (5.0 W m$^{-1}$ K$^{-1}$), another one with low thermal conductivity (2.5 W m$^{-1}$ K$^{-1}$) was also investigated. Thus, six simulations with corresponding model settings in Table 2 were projected with a long period from 1980 to 2100. Besides, similar simulations were also conducted for comparison on hydraulic and thermal pattern in 2008 by replacing the meteorological forcing with the observed air temperature and precipitation.

The atmospheric forcing used in this study is produced from the fifth Coupled Model Inter-comparison Project database of GCM output (CMIP5). The projected climate scenario of the Representative Concentration Pathway 8.5 (RCP8.5) was dynamically downscaled using the CanESM2/CGCM4 Model (Verseghy, 1991), which corresponds to a usual warming scenario with 8.5 W m$^{-2}$ forcing by 2100. Figure 2 provides the projected changes in mean annual air temperature (MAAT) and annual total precipitation from 1900 to 2100. Generally, a pronounced increase in air temperature started in 1980s and there is also a noticeable change in precipitation. These features are generally consistent with the regional trend of air temperature and precipitation obtained from local observations (Guo and Wang, 2013, Hu et al., 2014). Considering a quick



warming in permafrost during the past few decades (Cheng and Wu, 2007), a reasonable hypothesis is to presume the climate as steady for the 80 years before 1980s. Accordingly, we assume that the thin permafrost around 1980s was in pseudo equilibrium. Therefore, the model was spun up with the atmospheric forcing by using a repeated 10-year period from 1970 to 1979 that keep the mean annual soil temperature change less than 0.01˚C in all soil layers. The initial condition for the spin-up was a constant ground temperature of -0.5˚C and a water pressure in static equilibrium with a water table at 1.0 m below ground.

## 3 Results and Discussion

### 3.1 Relationship between air temperature and near-surface soil temperature

Figure 3 shows the relationship between daily mean air temperature and near-surface soil temperature (5 cm below ground surface) over the period of 2006-2014. A few sporadic outliers are related to abrupt cold weather, e.g., summer/autumn freezing. The well fitted linear fit indicates that the freeze-thaw process does not exert significant impact on heat transfer, which means a small change in seasonal thermal properties. This might be attributed to the seasonal liquid water change. In addition, the average temperature difference was about 5.0˚C, whereas the mean annual air temperature was even higher than -5.0˚C. Thus, the mean annual near-surface temperature should fluctuate around 0˚C.

### 3.2 Surface and thermal offsets

Figure 4 shows the thermal profiles measured over the period of 2007-2013. It covers the mean annual temperature data from 1.5 m above ground surface to 2.18 m in subsurface. The interaction between the lower atmosphere and permafrost can be characterized with surface and thermal offsets (Smith and Riseborough, 2002). Limited by available measurements, surface offsets were approximately estimated from the difference between mean annual near-surface (10 cm) temperature (MAGST) and MAAT, and the thermal offsets were calculated from the difference between mean annual temperature (MAT) close to the permafrost table (2.18 m) and MAGT. Here we should mention that these thermal offsets were overestimated slightly due to the used bottom MAT, which is not exactly at the permafrost table. Calculations show that the surface offsets in 2008, 2009 and 2013 were 4.40˚C, 3.78˚C and 4.30˚C, respectively. These values indicate a weak coupling between the lower atmosphere and ground surface. While the thermal offsets changed from positive values, i.e. 0.47˚C (2007) and 0.33˚C (2008) to negative ones, -0.18˚C (2009) and -0.15˚C (2013). Surprisingly, the positive thermal offsets occurred in colder weather conditions. Similar phenomenon has been found in a nearby region by Lin et al. (2015). This result seems to conflict that permafrost commonly exists a negative thermal offset (Smith and Riseborough, 2002). This might be related to the unique hydraulic and thermal dynamics in the active layer, which can cause a reversed seasonally variable thermal conductivity.





### 3.3 Pattern of hydraulic and thermal regimes in the stratified active layer

Figure 5 shows a typical annual evolution of the active layer to the weather condition (MAAT: -4.40˚C and total rain: 316 mm) in 2008. Figure 5a reflects a typical climate regime with dominant rainfall in the rainy season from June to September on the northeastern QTP. Note that the precipitation measurements mainly include rainfall, and light snowfall was detected

by an acoustic sensor to measure distance change. Figure 5b shows the active layer during an annual freeze-thaw cycle. The distribution of liquid soil water indicates that a large amount of suprapermafrost groundwater existed during the period from late May to the end of January with a maximum thickness of this saturated layer in excess of 1 m. The groundwater table roughly fluctuated around 1.0 m below the ground surface, and was mainly recharged by rainfall infiltration during the thawing period (late April to late October).

A noteworthy pattern of the water content distribution is the dry inter-layer around 0.7 m. This layer was occasionally wetted by rain infiltration during the rainy reason, but else was rather dry, as was also the case during the freezing period. This situation results from a fine-textured and less permeable layer overlying a coarse-textured one. We anticipate from the seasonal contrast in liquid soil water (reduction from the thawing period to the freezing period) in this dry layer will modify the seasonal thermal properties of the active layer.

### 15 3.4 Effect of seasonal liquid water content reduction on the ratio of thermal conductivity

For active layers with mineral soils, the ratio of thermal conductivity in the thawed and frozen states ($\lambda_t/\lambda_f$) is assumed to be less than or equal to one (Riseborough and Smith, 1998). However, the factor of seasonal liquid water content reduction is not negligible at our study site. Figure 6a compares the change in total liquid water content between summer and winter in 2008. The maximum seasonal liquid water content reduction occurred around 0.7 m in the active layer. Assuming a thermal

conductivity of 5.0 W m$^{-1}$ K$^{-1}$ for the sand with high content of quartz, the thermal conductivities at different depths were calculated with Equation (1) in Fig. 6b. There are two locations with $\lambda_t > \lambda_f$ at 0.65 and 0.89 m depth. Significant reduction of the liquid water content also occurred at other depths, all accompanied by corresponding reductions of $\lambda_f$.

In order to exceed the ratio of 1, the seasonal liquid water content has to fall below a certain threshold, which depends on soil thermal conductivity and water content in thawed state. For instance, the soils with high thermal conductivity of soil

matrix will need larger liquid water content reduction than that of the soils with small thermal conductivity of soil matrix. However, given the same amount of liquid water content reduction, the soils with a low soil water content in thawed state will be prone to reach a ratio over 1. Generally, the soil water content condition in thawed state depends on soil type and soil structure. Considering the unique precipitation characteristics on the QTP, seasonal liquid water content reduction is common in this kind of permafrost regions. Unfortunately, the role of soil architecture in thermal conductivity

parameterization is rarely addressed to date.



### 3.5 Comparison of observed and simulated permafrost warming rates

In this section, the model is validated by comparing with the observations. Figure 7 compares the observed and CMIP5-projected air temperature over the period of 2006-2014. Generally, the patterns were very similar in Fig. 7a, but there was a daily averaged up-shift of 0.96°C of the projected values according to the linear regression of all available values (Fig. 7b). Since there were several data gaps in the observed air temperature, it is difficult to derive a trend of the measured MAAT change in Fig. 7c. However, a linear-fitted warming rate of 0.07°C yr$^{-1}$ of the projected MAAT from 2006 to 2100 can be derived from Fig. 2a, it is higher than that of the reported warming trend (0.02°C yr$^{-1}$) between 2003 and 2012 for a reference station (Beiluhe) in the same region (Wu et al., 2015).

Figure 8 compares the observed and simulated permafrost temperature changes over the period from 2006 to 2014. The measured temperature $T_{obs,06}$ was taken from a shallow borehole on August 30, 2006 and $T_{obs,14}$ was from a nearby deep borehole on February 22, 2014. Here we assume that permafrost ground temperature distribution was roughly uniform within a small area (30 m×30 m) due to similar surface and subsurface properties. Thus, permafrost warming can be deduced from ground temperature change from the boreholes. Considering the small annual fluctuation of ground temperature change at the depth of 10 m, two corresponding measurements irregularly conducted once a year in 2006 and 2014 can roughly provide a permafrost warming rate of 0.05±0.1°C yr$^{-1}$, concerning the uncertainty of annual fluctuation as simulated ones shown in Fig. 8. By assuming a constant warming rate in permafrost temperature at the depth of about 10 m, a value of 0.02°C yr$^{-1}$ was calculated from $T_{sim,06}$ and $T_{sim,14}$. This is just about half of the observed one. Compared to the projected warming rate of air temperature at the same period (0.07°C yr$^{-1}$), the simulated warming rate is underestimated.

The evident discrepancy between observed and simulated permafrost warming rate is mainly attributed to the following three factors. First, snow process is not represented reasonably in the model due to the limitation of the meteorological forcing. Permafrost is extremely sensitive to snow cover, which has a much higher albedo (> 0.9) than regular ground surface (0.1-0.4). Field observations show that snow cover only lasts a few weeks in pre/post winter, and the missing snow cover is mainly caused by evaporation and sublimation during the diurnal thawing. However, the projected meteorological data are in daily resolution and the precipitation is also not well accurate in general. Second, the atmospheric forcing was down-scaled from large scale climate modeling, and it differs from site observations. Particularly, the observed increasing rainfall but less snowfall is not well predicted in the projected meteorological forcing. Third, some simplifications like constant surface albedo and flat ground without lateral flow, as well as empirical parameters might also influence the simulations. Nevertheless, the model can reasonably mimic hydraulic and thermal regime and current permafrost thermal status, and it can help us to investigate the effect of the stratified active layer on permafrost warming.

### 3.6 Role of the stratified thick active layer in permafrost warming

In this section, the effect of the stratified active layer on permafrost warming is validated with modeling, and subsurface controlling factors in the active layer including soil architecture and thermal conductivity of soil matrix are examined with





thermal offset and permafrost temperature. Subsection 3.6.1 demonstrates the simulated hydraulic and thermal pattern in the active layers, and subsections 3.6.2 and 3.6.3 present the evolution of the unique thermal offset and permafrost temperature, respectively.

### 3.6.1 Simulated hydraulic and thermal pattern in the active layers

Given a typical observed meteorological forcing in 2008, the simulated hydraulic and thermal patterns of the active layers with different soil architectures (A1, A2 and A3) in 2008 are shown in Fig. 9. Generally, the thermal conductivity of the soil matrix dominates the active layer thickness, regardless of the soil architecture. However, contrast soil hydraulic patterns are controlled by hydraulic properties, mainly soil architectures here, and also influence the active layer thickness. Notice that the order of the maximum thawing depth is A3 > A1 > A2, which is similar in both columns. In addition, the one with

realistic soil architecture A3 in Fig. 9c presents a similar hydraulic and thermal pattern as the one observed (Fig. 9c′), and this shows also that the model can reasonably capture the hydraulic and thermal dynamics in the active layer. However, the simulated downward thawing in early summer is slower than the observed one. This is mainly related to the underestimated permafrost temperature in the modeling.

### 3.6.2 Evolution of the unique thermal offset

Based on the above different hydraulic and thermal patterns we investigated the impact of soil architecture on the warming of the underlying permafrost by using the thermal offset. Figure 10 shows the evolution of the thermal offset with the change of MAAT, the latter the result of climate warming. The thermal offset is calculated as the annual temperature difference ($T_{top}$ - $T_{0.1m}$) between the top of the permafrost table and the near-surface (0.1 m), and disappears till talik present, which disconnects the permafrost from the seasonal frost layer. Generally, all the thermal offsets decreases with increasing MAAT.

However, the thermal offsets of A3 in Fig. 10a and 10b are both positive at the beginning of the simulation, when the permafrost is close to thermal equilibrium. Whereas, the thermal offset of A2 is always negative and the thermal offset of A1 is in between A2 and A3.

Apparently, the active layer A3 here does not conform to the expectation of a negative thermal offset, i.e., TTOP < MAGST due to $\lambda_f > \lambda_t$ (Smith and Riseborough, 2002). The positive values close to the initial equilibrium state in Fig. 10

indicate that the permafrost at the simulated condition does not fit the concept of thermal offset. The schematic mean annual ground temperature profile is more close to the one described by Brown (1970) other than the one suggested by Smith and Riseborough (2002) in Fig. 11. The reversed thermal offset at equilibrium state was mainly led by the high ratio of $\lambda_t/\lambda_f$ around 1 via seasonal liquid water content reduction. Whereas, the subsequent "normal" thermal offsets were not caused by $\lambda_f > \lambda_t$ but by non-equilibrium effects. This is corroborated by the observed thermal offsets (Fig. 4) that positive values

occurred in 2007 and 2008 and they decreased to negative in 2009 and 2013. Therefore, the concept of the normal offset is not suitable for the studied case, and the plausible "normal" thermal offset might not necessarily be attributed to $\lambda_f > \lambda_t$, but to permafrost degradation.





### 3.6.3 Subsurface factors controlling the permafrost warming rate

Apart from the change rate of climatic forcing, the evolution of thermal regime in active layers is also not negligible for permafrost warming, and it is related to subsurface factors like thermal conductivity of soil matrix and soil hydraulic properties. In particular, the thermal regime in the studied thick active layer with unique soil architecture strongly relies on the hydraulic regime. Given three different soil architectures and two different thermal conductivities of soil matrix, the differences of the permafrost warming are shown in Fig. 12. Generally, the permafrost with a higher thermal conductivity of the matrix (left column of Fig. 12) degrades faster than that in the right column with a lower thermal conductivity, and the influence of soil architecture in the left column is negligible. In contrast to, the role of soil architecture emerges in the right column, and the stratified active layer (A3) leads to a faster permafrost warming rate. This is mainly attributed to the effect of seasonal liquid water content change on seasonal variation of thermal conductivity. For the active layer with a high $\lambda_m$, the ratio of seasonal thermal conductivity is always close to 1.0, and the impact of seasonal liquid water content reduction is rather weak. However, for the active layer with a low $\lambda_m$, the ratio of seasonal thermal conductivity depends strongly on the seasonal liquid water content reduction. Besides, soil architecture with low soil water content will have a high ratio of seasonal thermal conductivity, given the same amount of seasonal liquid water content reduction.

### 4 Conclusions

In summary, this study presented an interesting case to show the effects of stratified active layers with a high ratio of seasonal thermal conductivity, $\lambda_t/\lambda_f \geq 1.0$, on permafrost warming on the QTP. Combining with intensive observations of soil hydraulic and thermal dynamics in the active layer and weather over a period of almost 9 years, as well as corresponding numerical simulations, key findings are listed in the following:

(1) An extraordinary permafrost warming rate (> 0.5°C per decade) was found at the study site with sparsely vegetation and annual precipitation 300-400 mm. Apart from the climate drivers and the unusual high geothermal flux from the bottom, a high ratio of seasonal thermal conductivity in the stratified active layer is indispensable in regulating the interaction between climate and permafrost.

(2) Observation and simulation suggest that the concept of the thermal offset proposed by Smith and Riseborough (2002) is not suitable for the studied permafrost on the QTP. In contrast to the normal thermal offset caused by the seasonally variable thermal conductivity, a reversed thermal offset at equilibrium state is formed due to the remarkable high ratio of seasonal thermal conductivity, namely close 1.0 or even higher, given such a weather pattern and soil properties.

(3) Furthermore, the specific soil architecture plays a non-negligible role in forming a dry inter-layer while facilitating to raise the ratio $\lambda_t/\lambda_f$, and resulting in a higher permafrost warming rate than the active layers with uniform soils.

Considering the importance of rainfall in the mechanism of the hydraulic and thermal dynamics in the active layers, there is no doubt that permafrost warming would be influenced by the increasing precipitation in recent years and in future on the QTP. Consequently, soil hydraulic properties, particularly soil architecture become more and more important for the thermal




conductivity parameterization in land surface and permafrost modeling. Particularly, the empirical permafrost models using a ratio of seasonal thermal conductivity smaller than 1.0 might underestimate the effect of climate warming. However, this study is mainly based on a specific site. More field investigations are required to reveal the regional difference in permafrost degradation over the QTP.

*Acknowledgements.* We acknowledge Dr. Yanhui You for field data collection and Dr. Liang Chen for processing climate data. This research was funded in part by the National Natural Science Foundation of China (Grant No. 41171059).



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



**Table 1.** Soil properties for the actual soil profile. $K_s$: saturated hydraulic conductivity; $\alpha$ and $n$: van Genuchten parameters; $\theta_r$ and $\theta_s$: residual and saturated soil water content, respectively; $\lambda_m$: thermal conductivity of soil matrix; C: thermal capacity.

| description | layering / m | 0-3.0 I | 0-3.0 II | 3.0-30 |
|---|---|---|---|---|
| Soil texture % | sand | 66.3 | 92.2 | - |
| | silt | 12.0 | 3.8 | - |
| | clay | 21.7 | 4.0 | - |
| Hydraulic properties | $K_s$ / m d$^{-1}$ | 0.19 | 4.68 | $2.2\times10^{-3}$ |
| | $\alpha$ / cm$^{-1}$ | 0.03 | 0.03 | 0.01 |
| | $n$ / - | 1.33 | 2.85 | 1.5 |
| | $\theta_r$ / m$^3$ m$^{-3}$ | 0.06 | 0.05 | 0.10 |
| | $\theta_s$ / m$^3$ m$^{-3}$ | 0.38 | 0.38 | 0.2 |
| Thermal properties | $\lambda_m$ / W m$^{-1}$ K$^{-1}$ | 5.0 | | 2.0 |
| | C / J m$^{-3}$ K$^{-1}$ | $2\times10^6$ | | $2\times10^6$ |



**Table 2.** Combinations of soil architecture and thermal conductivity of soil matrix for the shallow layer (0-3.0 m) for the six simulations. A1, A2 and A3 stand for three types of soil architecture for the shallow layer.

| Simulations | 1 | 2 | 3 | 4 | 5 | 6 |
|---|---|---|---|---|---|---|
| Architecture | A1 | A2 | A3 | A1 | A2 | A3 |
| $\lambda_m$ / W m$^{-1}$ K$^{-1}$ | 5.0 | 5.0 | 5.0 | 2.5 | 2.5 | 2.5 |



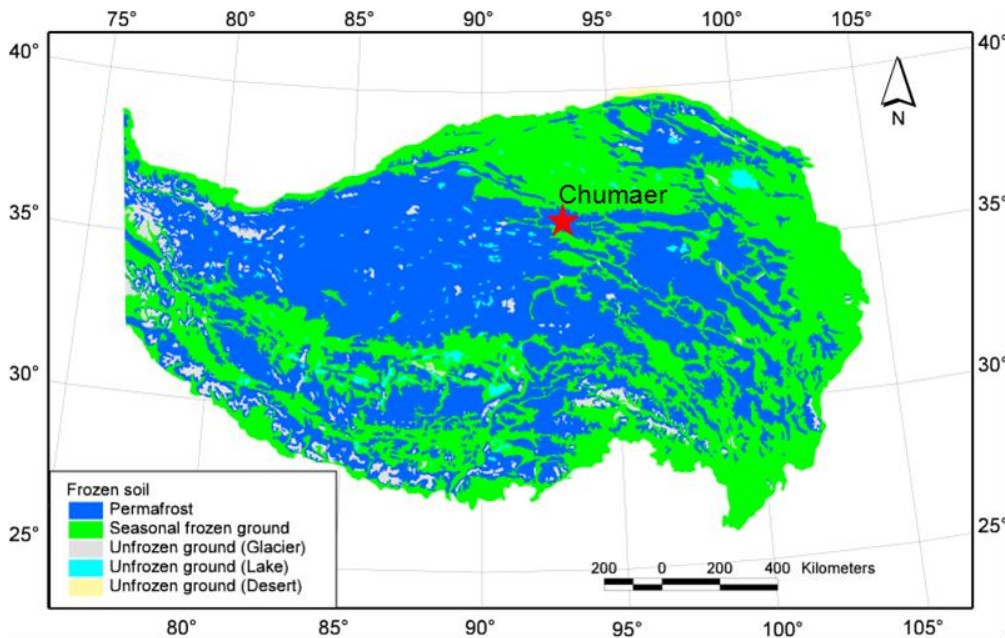

**Figure 1.** Study site location and permafrost distribution on the Qinghai-Tibet Plateau. The background map is the permafrost classes from Li and Cheng (1996).





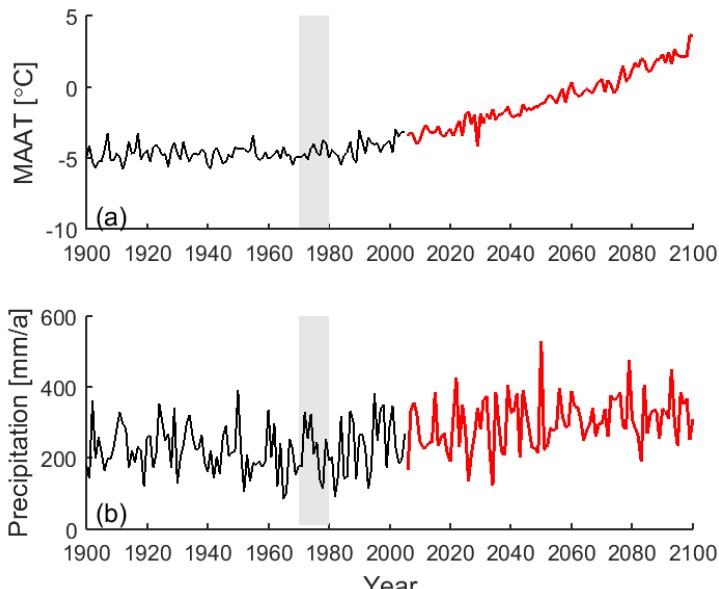

**Figure 2.** Time series of projected mean annual air temperature (MAAT) (a) and annual total precipitation (b) since 1900 to 2100. The black section presents historical data, and the red section presents projected data. The period in shadow (1970-1979) was repeatedly used for spin-up.





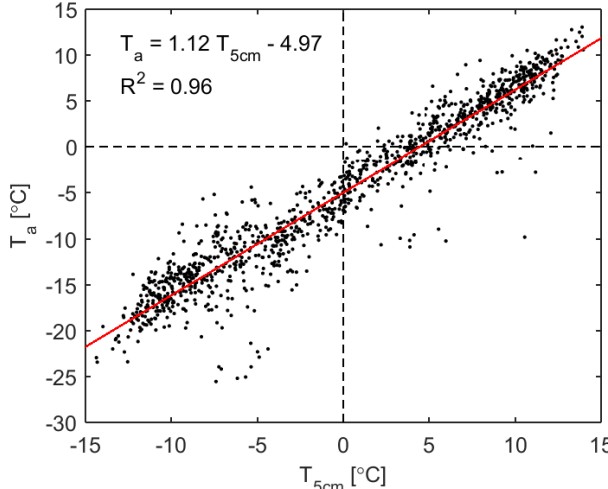

**Figure 3.** Relationship between the daily mean air temperature ($T_a$) and near-surface soil temperature ($T_{5cm}$, measured at 5 cm below the ground surface) over the period from 2006 to 2014.



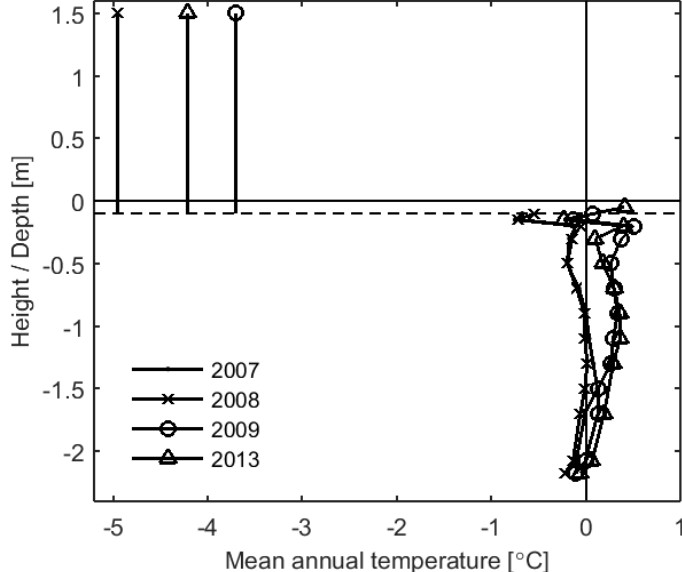

**Figure 4.** All available thermal profiles from 2007 to 2013. The temperatures at 1.5 m are the mean annual air temperatures (MAAT). Note: missing years were caused by data gaps.





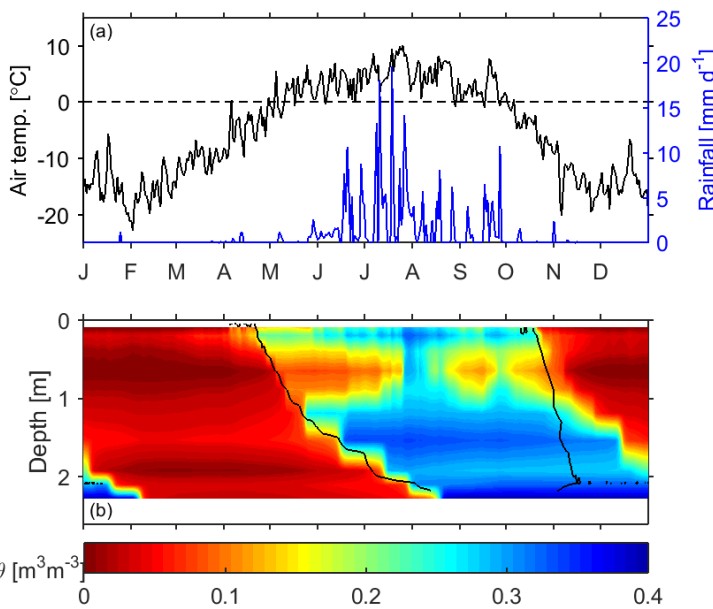

**Figure 5.** Typical dynamics of the active layer during an annual cycle (data from 2008). (a) Daily mean air temperature and daily rainfall. (b) Liquid water content (colors) and 0°C-isotherm (black line).



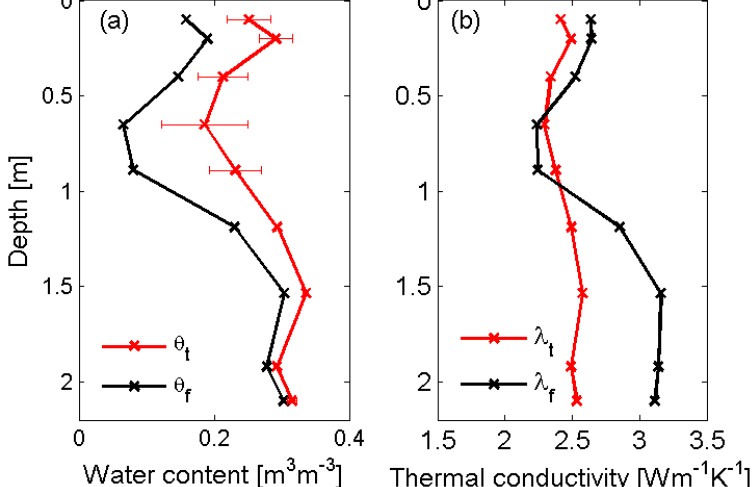

**Figure 6.** Seasonal change in soil water content (a) and thermal conductivity (b) in the thick active layer in 2008. $\theta_t$ and $\theta_f$ are the mean total water content during the summer period and the winter period, respectively, and $\lambda_t$ and $\lambda_f$ are the corresponding thermal conductivities.





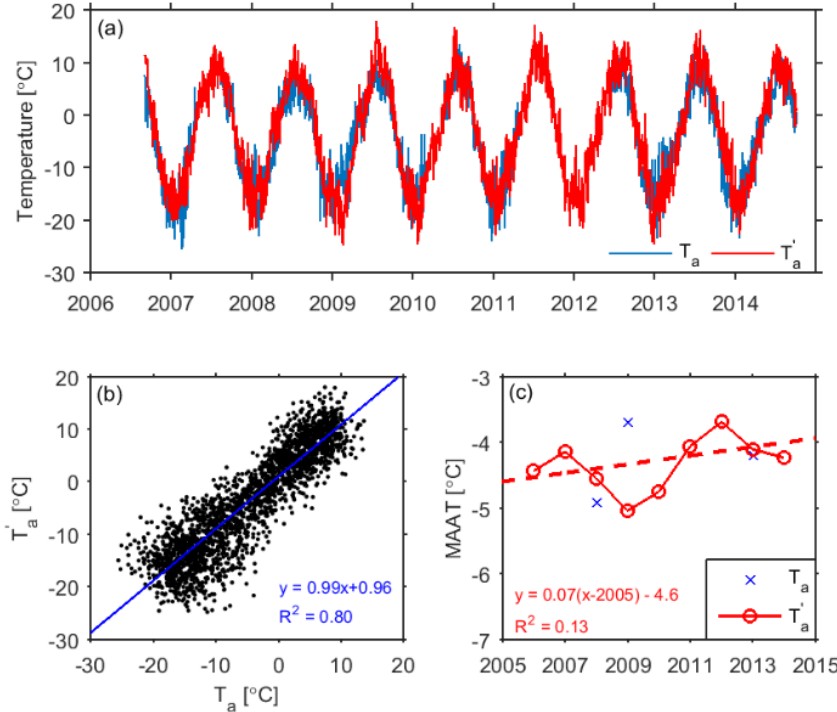

**Figure 7.** Air temperature change over the period of 2006-2014. (a) Comparison on observed and projected daily averaged air temperature ($T_a$ and $T_a^{'}$); (b) Relationship between $T_a$ and $T_a^{'}$ with a linear regression; (c) Comparison on the observed and projected annual mean air temperature (MAAT) after correction with (b). A linear-fitted (dashed line) warming rate of 0.07°C yr$^{-1}$ is derived from the projected one.





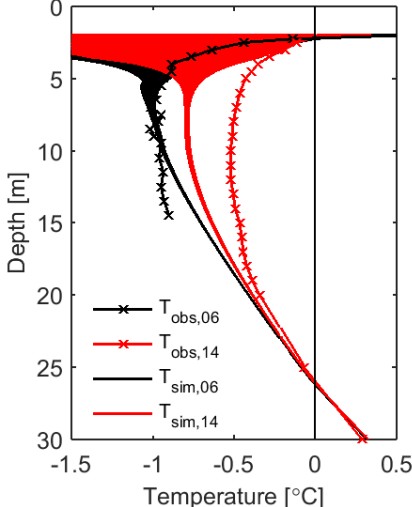

**Figure 8.** Comparison on observed and simulated permafrost temperature changes at the Chumaer site over the period from 2006 to 2014. The measured temperatures $T_{obs,06}$ and $T_{obs,14}$ were taken from a shallow borehole and a nearby deep borehole on August 30, 2006 and on February 22, 2014, respectively, while the simulated ones $T_{sim,06}$ and $T_{sim,14}$ show all values of the corresponding years.





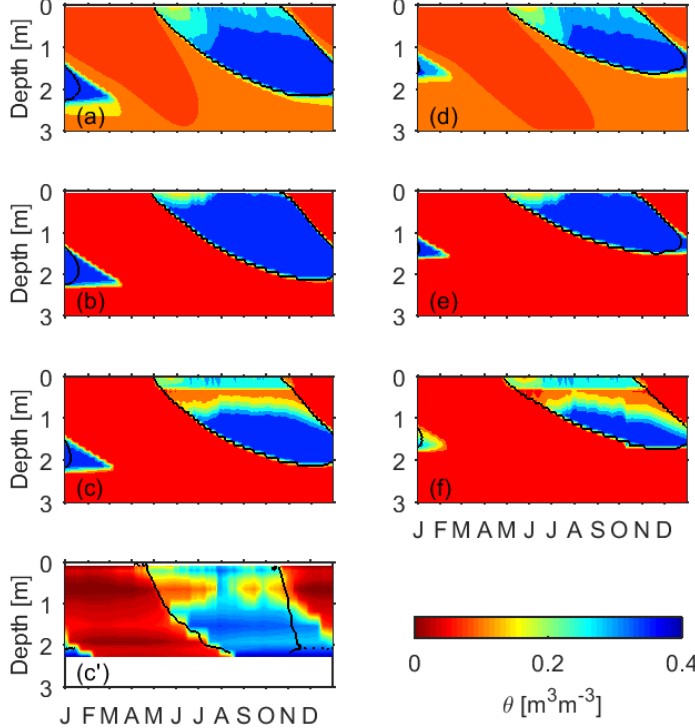

**Figure 9.** Comparison of the simulated unfrozen soil water content (color) and the 0°C-isotherm (black line) in the active layer with different soil architectures and thermal conductivities in 2008. The rows correspond to different architectures of the soil: (a, d) single fine-grained layer (A1), (b, e) single coarse-grained layer (A2), (c, f) two layer structure (A3). The columns correspond to different thermal conductivities of the soil matrix: 5.0 W m$^{-1}$ K$^{-1}$ (left) and 2.5 W m$^{-1}$ K$^{-1}$ (right). (c′) Observed case (Same as Fig. 5b).





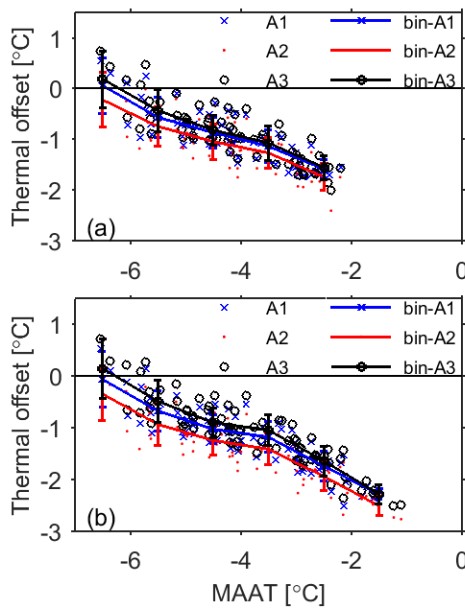

**Figure 10.** Thermal offset as a function of mean annual air temperature (MAAT) obtained from the simulations for the period 1980-2100 of the three soil architectures A1, A2 and A3. The lines connect the means of 1.0°C-bins, the bars indicate the corresponding standard deviations. The upper frame is for $\lambda_m = 5.0$ W m$^{-1}$ K$^{-1}$, the lower one for $\lambda_m = 2.5$ W m$^{-1}$ K$^{-1}$.



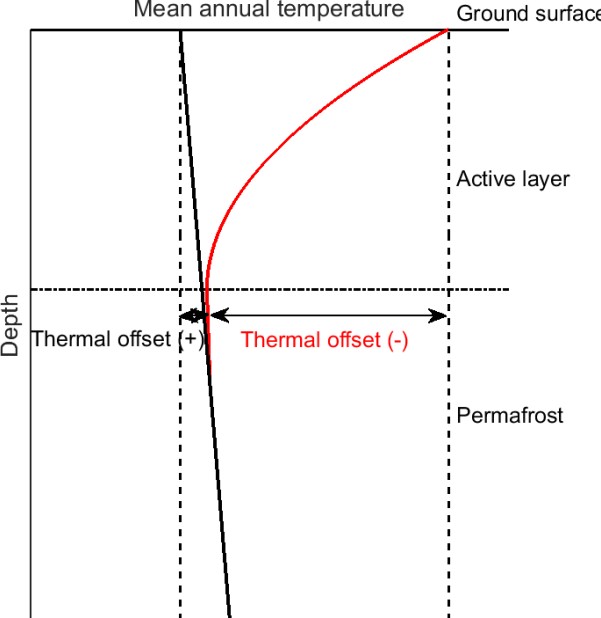

**Figure 11.** Schematic mean annual ground temperature for two types of permafrost. Black curve: the studied permafrost with positive thermal offset; red curve: common permafrost with negative thermal offset.





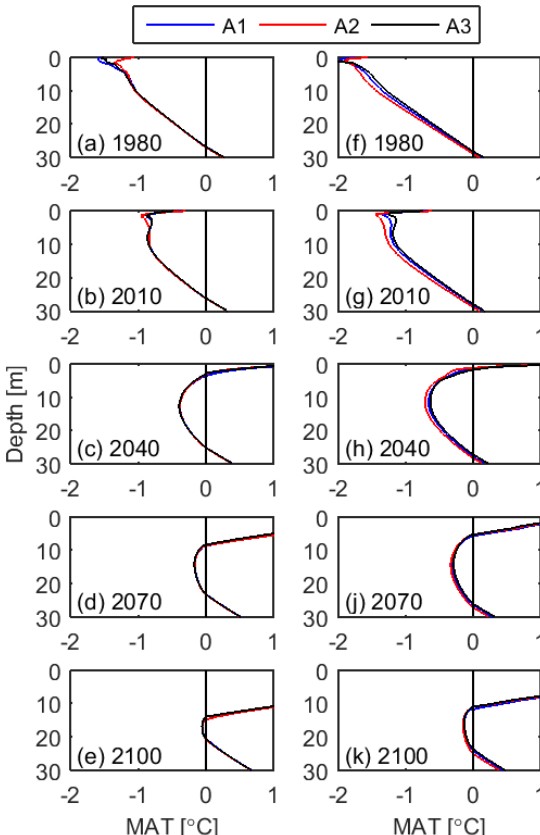

**Figure 12.** Comparison of the influence of soil architecture and thermal conductivity of soil matrix on permafrost degradation with simulations over the period from 1980 to 2100. (a) - (e) Annual mean ground temperature (MAT) of A1, A2 and A3 with a high thermal conductivity of soil matrix (5.0 W m$^{-1}$ K$^{-1}$) at selected years; (f) - (j) the same as (a) - (e) but with low thermal conductivity of soil matrix (2.5 W m$^{-1}$ K$^{-1}$).