# Peer review of "Effects of Stratified Active Layers on High-Altitude Permafrost Warming: A Case Study on the Qinghai-Tibet Plateau"

_The Cryosphere, 2015_

## Referee Comment (RC1) · Anonymous Referee #1 · 19 Jan 2016

Review of Pan et al., 2016 HESSD

Pan et al. demonstrate that the hydrologic (or hydrogeologic) properties and conditions of the active layer influenced the active layer thermal regime and hence the rate of permafrost warming in response to climate change. In particular, they demonstrate that the active layer hydrologic properties can cause a thermal offset that is opposite in magnitude than what is normally observed (i.e. their TTOP > MAGST) and that not considering this in permafrost models can lead to underestimating future permafrost thaw. I think that the study is of interest and may warrant publication eventually. The authors need to make a number of points more clearly. I also question their modeling results/interpretation.
[Figure]

Major comments

1. The authors need to be clearer regarding what causes thermal offsets. They mention vague processes like 'seasonal variability in thermal conductivity' (P2,L15-20; Section 3.2, Section 3.6.2) but they can be much more explicit. It is not complicated. If the moisture content is the same throughout the year, thermal conductivity is higher in winter than in summer because pore ice has about four times the thermal conductivity of pore water. Thus heat transfer will be more efficient in winter than in summer causing a thermal offset (TTOP is colder than MAGST). In order for the sign of this offset to be switched, heat transfer must be more efficient in the summer. This means that moisture content must be considerably higher in the summer than in the winter. For example, if the pore ice has a thermal conductivity roughly four times the thermal conductivity of pore water (and assuming that air has a negligible thermal conductivity, which it pretty much does), than a homogeneous medium would need over four times the moisture content in the summer compared to the winter to have more efficient summer heat transfer. Correct? I feel like these simple facts are obfuscated in the manuscript. This should especially be addressed at the end of section 3.6.2. The authors suggest that climate warming and permafrost disequilibrium can cause the offset observed by Smith and Riseborough and others. They should explain, phenomenologically, why this would be the case. I would have thought that the lag between surface and subsurface warming would actually make the thermal offset more negative with time under extreme warming. So the results are unintuitive in my opinion. This could potentially make them more interesting, but they have to be explain from a physical perspective.

2. The authors jump right into permafrost thaw in the introduction without any mention of why it is important. I understand that this is a cold regions journal, but some context would be nice. I think the authors should highlight that permafrost thaw has considerable implications for surface and subsurface hydrologic routing (e.g. Kurylyk et al., 2014 ESR), geotechnical failures (Harris et al., 2009, already cited), and carbon dioxide and methane release (Schuur et al., 2015). Those papers are broad review

papers on permafrost thaw impacts.

Kurylyk BL et al. 2014. Climate change impacts on groundwater and soil temperatures in cold and temperate regions: Implications, mathematical theory, and emerging simulation tools. Earth Sci. Rev. 138, 313-334.

Schurr et al. 2015. Climate change and the permafrost carbon feedback. Nature 520: 171-179.

3. Section 3.1. I don't understand how there could be a 5°C difference between the mean air temperature and mean surface (or very close to surface) temperature. Such differences are commonly observed (e.g. Zhang et al., 2005) but only in regions that experience deep snowpack (Zhang, 2005). This seems like a very high difference without considerable snowpack. Could the authors explain?

Zhang et al. 2005. Soil temperature in Canada during the twentieth century: complex response to atmospheric climate change. J. Geophys. Res. 110 (D03112)

Zhang T. 2005. Influence of the seasonal snow cover on the ground thermal regime: an overview. Rev. Geophys. 43 (4).

4. Section 2.4.1. Where does the water go in the model if no lateral flows are allowed (P4, L11) and no vertical drainage is allowed out of the bottom (P4, L23)? Are ET and P presumed to be balanced? This is very confusing. Also, P4, L26 implies that the thermal conductivity and porosity are used to computer the geothermal flux, when it is really the conductivity times the gradient. Also, in this paragraph and in many other cases through the manuscript, the authors use the term 'soil matrix' to refer to the solid particles. Sometimes matrix (in the context of thermal conductivity) means the matrix of water and solid grains. I recommend that here and elsewhere the authors change the terminology from soil matrix to 'soil particles' or 'soil grains'.

5. The modeling results (simulated vs. observed, Fig. 8) are not good. This modeling exercise certainly did not 'validate' (P8, L2) their model. Observed warming is about

twice the modeled rate. The authors propose that this is due to (1) not accounting for snow, (2) difference in simulated and observed climate data, and (3) model assumptions. Okay I can buy that. But why not examine (1) by comparing measured and simulated surface temperatures? In other words, if the problem with the model occurs at the atmosphere-soil interface, then you can easily demonstrate that by comparing the measured and simulated temperature at this point. Also, why did the authors use climate model data for a period when they had site data? This makes no sense at all. Just use the climate data for the future period, not for the model performance assessment period. So I believe the authors can easily test (1) and (2) above.

6. Section 3.6.3 and Figure 12, The differences in the thermal regimes for the three soil architectures (A1, A2, and A3) seems rather minor in my opinion (for both the left and right columns in Figure 12). Since this is a major point of the paper, I'm left wondering, 'what's the point'? The series are virtually indistinguishable from 2040 onwards.

Minor comments

Title, delete 'the' as it is not needed and sounds funny

P1, L16-17, delete 'with thickness larger than some 1.5 m' and insert '(>1.5 m) after 'thick'

P1, L17, 'It is additionally furthered' is unclear. This should be something like 'The conductivity ratio can be further increased' or something like this

P1, L23, 'high mountains' should be 'alpine regions'. The Alps is not a mountain, for example.

P1, L26, delete 'the' after 'Generally' and 'the' after 'than'

P1, L28, 'Their response'....whose response?

The last sentence on P1 should be moved before the sentence beginning with 'For instance' The first paragraph reads a bit like a Wu et al. fan club press release. I think

it would be better to incorporate some of the implications of permafrost thaw in this paragraph (see major comment above)

P2, L3, The reported permafrost warming rate is half of the air temperature warming rate (we would expect it to be lower, so that is fine); however, the paragraph reads as if the permafrost warming rate is higher. I'm not convinced that it makes sense to compare the permafrost warming to air temperature warming over a 13 year period (2002-2014). The subsurface warming rate is lagged (and typically damped) in response to a surface (or atmospheric) warming period. The lag is not that important when you are talking about a 100 year period, but it certainly is over 13 year period.

P2, L8, insert 'the' before 'atmosphere'

P2, L11, I'm confused by the comment regarding diurnal forcing and freeze-thaw days. Permafrost is not really diurnally forced.

P2, somewhere the authors could consider citing Hayashi et al. 2007 who proposed a Stefan type algorithm to deal with the problem the paper focuses on (i.e. changing moisture content through the season).

Hayashi et al. 2007. A simple heat conduction method for simulating the frost-table depth in hydrological models. Hydrol. Process. 21(19)

P2, L22, Delete 'the' before 'permafrost'

P2, L23 Delete 'a' before 'recent'

P2, L25, I don't think diagnose is the right word here....maybe characterize?

P3, L12. If the permafrost is 25 m, than the soil at a depth of 10 m must be permafrost. So of course, the temperature would have to be less than 1.0°C. In fact, it would have to be less than 0°C or it is not permafrost.

Heading for section 2.2 contains an extra 'Subsection'

P3, L27, Delete 'the' before 'GEOtop'. Also this sentence would read much better if 'surface and soil, and the soil freezing' were replaced with 'surface and soil as well as the soil freezing and thawing'. Otherwise it sounds like freezing and thawing is included in the list containing atmosphere, surface, and soil.

P3, L29, Change 'allows to simulate' to 'simulates'

P3, L30, Delete 'the' before 'complex'

P4, L2, the concept of relating the soil freezing curve and soil drying curve is quite foreign to most permafrost scientists. Consider citing the review on this topic.

Kurylyk and Watanabe. 2013. The mathematical representation of freezing and thawing processes in variably-saturated, non-deformable soils. Adv. Wat. Res. 60, 160-177

P4, L2, change 'allows' to 'enables the user' P4, L11 delete 'given as'

P4, L15, change 'with a high resolution in size of 10 cm. . .and was gradually reduced' to something like 'with elements with a height of 10 cm. . .and reducing to. . .' or something like that

P5, L14-16, This is a fragment and confusing

P5, L18, insert 'the' before 'active layer'. Insert 'the' after with

P5, L24, change 'on' to 'of the'

P5, L27. There should be an appropriate citation for CMIP5. If I remember correctly, there is a brief paper published describing the dataset

P5, L32, change 'a quick' to 'the rapid'

P6, L3. I'm curious how many 10 year periods were run for the spin up (i.e. how many cycles). This should be mentioned.

P6, L11, delete 'well-fitted'

P6, L11, insert 'in the very shallow subsurface' after 'heat transfer'

P6, L16, insert 'mean annual' before 'thermal profiles. Delete '. It covers the mean annual temperature data', i.e. combine first two sentences into 1.

P6, L21. MAGT should be MAGST shouldn't it?

P6, L26, insert 'with the fact' after conflict

P6, L27, change 'exists' to 'exhibits'

P7, L11, change 'else' to 'otherwise'

P7, L12, change 'from' to 'that'

Last sentence in P7 sounds like it should be in introduction not $\frac{3}{4}$ of the way through the paper

P8, L7, Delete 'it is higher…..Wu et al. (2015)'. This is not relevant given how different the periods are.

P8, L31, 'validated' should be 'investigated' or something like this

P9, L7, change 'contrast' to 'the'

P9, L13, 'underestimated permafrost temperature' is not really a good physical explanation for why thawing is slower in the model than in observations. Of course, this is caused by underestimated permafrost temperature, but the question that should be addressed is 'why is the permafrost temperature underestimated?'

P9, L18, I'm confused by the statement 'and disappears till talik present'

P9, L26, 'more close' should be 'closer'

P10, L8, delete 'to'

P10, L17-19, this is a fragment

P10, L20, is this 'extraordinary permafrost warming rate' referring to observed or simulated warming?

P10, L25-27. This sentence seems to contradict itself (although I know what the authors mean): 'In contrast to the normal offset caused by the seasonally variable thermal conductivity, a reversed thermal offset at equilibrium state is formed due to the remarkable high ratio of seasonal thermal conductivity

Figure 4 – different colours for the series would be helpful (after all TC is all online anyway)

Figure 6 caption. The thermal conductivities in (b) are calculated via Eq. (1) right? If so, this should be stated in the caption. Also, how is the ice content obtained for this equation? Somewhere it is stated that the moisture content is assumed to stay the same in the winter. So then the ice is calculated as the total minus liquid?

Figure 7 caption: change 'on' to 'of' in both places.

Figure 9, 10, and 12. The authors should clearly highlight the differences between the left and right columns (Figure 9 and 12) and the top and bottom (Figure 10). I think it is better to label the figure panels rather than put this info in the caption. Otherwise the reader is scanning up and down.

Table 1. How was the solid particle thermal conductivity of 5 W/(m K) chosen? This is rather high for sand grains in my experience.
[Figure]

---

## Referee Comment (RC2) · Anonymous Referee #2 · 17 Feb 2016

**Review of the Manuscript : Effects of Stratified Active Layers on the High-Altitude Permafrost Warming: A Case Study on the Qinghai-Tibet Plateau**

**by X. Pan et al.**

**MS No.: tc-2015-201**

**General Comments**

The authors report about a specific phenomenon, namely a positive thermal offset monitored at a site in the Qinghai-Thibet Plateau. They explain the physical basis for this observation. Explanations are supported by modelling, observations, and are confronted to relevant scientific literature.

I congratulate the authors for the straightforwardness of their study, which reads well and leads to sharp conclusions. Plus, the topic is very accurate and the effect of stratified soils has been critically under-documented in the permafrost literature. Therefore the authors'findings are of major interest for this research community.

However, the other side of the straightforwardness is that the mentionned effect and its persistence over time should be supported by (more) relevant quantitative arguments (detailed exemples are given in the **Specific Comments** below). Also, the implications of the authors'findings at large scales or for other, possibly similar permafrost regions, and with respect to possibly changing precipitation patterns, could be more discussed. This would increase the paper's impact.

**I therefore recommend the paper for publication, pending the revisions detailled below.**

**Specific Comments**

➢ The formula for « thermal offset » and « surface offset » should be recall (in the introduction) for better clarity. In section 3.2, confusion is introduced about « thermal offset » : it was defined in the introduction as « TTOP – MAGST ». In section 3.2 it is approximated by « T(-2.18 m below surface) – MAGT » with MAGT quite different from MAGST. Please clarify.

➢ Section 2.2 is entitled : 2.2 Surface-subsurface monitoring scheme Subsection. « Scheme subsection » could be deleted from the title.

➢ The defined soil architectures in Section « 2.4.2 Simulation protocol » are not consistent with the caption of Fig 9 and the explanations of Section 3.6.1. Please make sure the Architecture definition is consistent in the whole document (maybe add a Table).

➢ P8 l 2, L31 : neither the model nor the effect are 'validated' in the current state of the paper. The comments below may give some sense to the validation of the effect through modelling.

➢ Concerning the local λt/ λf ratio : Year 2008 is used as an illustration of typical annual conditions. Given that ground temperature and soil water content are being measured at this site since 2006, stepping back from Year 2008 and bringing an interannual perspective would strenghen the paper's conclusion. I at least recommend a Table with the maximum λt/ λf value over the upper 2.18 m of the soil for each year with observations.

➢ Concerning the impact of the λt/ λf ratio on permafrost warming :
   o Fig 6 could provide the vertical profiles for λt and λf with λm=2.5 W/m/K, in support of the assessment : «*In order to exceed the ratio of 1, the seasonal liquid water content has to fall below a certain threshold, which depends on soil thermal conductivity and water content in thawed state. For instance, the soils with high thermal conductivity of soil matrix will need larger liquid water content reduction than that of the soils with small thermal conductivity of soil matrix.*»
   o A high λt/ λf ratio is advanced as an important argument for en enhanced permafrost warming rate at the observation site. However, Fig. 12 is the only illustration supporting this thesis (as modelling - Fig 8 - fails to reproduce the observed warming) ; it shows that permafrost warming rate is enhanced in the A3 configuration ; the authors explain that this is due to higher λt/ λf ratio, but this ratio is unfortunately never explicited. I highly recommand adding the mean interannual λt/ λf for each of the 10-year periods preceeding the selected years of Fig. 12, and for each soil architecture. This would make the paper's main argument less vague. This point is a **Major Comment**.
   o P 10 l 13 : the formulation could be improved (like : high -> higher)

➢ P5 l 30 and P8 l 24 : a crucial thing is to know whether the annual cycle of precipitation in the chosen downscaled projections, is still monsoon-like (as today) or shifts to different patterns in future climate. The authors mention that the projected rainfall may not be accurate. However, given the importance of the annual rainfall pattern on the site specific sub-surface thermal dynamics, more investigations on the projected precipitation pattern in the chosen downscaled climate product is needed, in support of the assessment of the impact of $\lambda t$/ $\lambda f$ on the warming. This point is a **Major Comment**.

➢ the implications of the authors'findings at large scales or for other, possibly similar permafrost regions, could be more discussed. Do the authors suspect that other sites could show similar caracteristics ?

**Technical Corrections**

- Very frequently the authors confuse « whereas » with « while » or « in the opposite ». (p4 l 23 ; p5 l 14 ; p6 l 13 and l 24 ; p9 l 28 ; …)
- P2 l 14 : basing -> based
- P3 l 15 : humility -> humidity
  P4 l 2 : incomplete sentence
  P5 line 13 to 16 : unclear, please reformulate
- P9 l 18 : till talik -> when talik

---

## Author Comment (AC1) · 18 Mar 2016

We thank the reviewer for her/his insight comments and suggestions, which make modeling results and interpretation clearer and conciser. We tried our best to implement the suggested changes. This includes the addition of figures, references, as well as wording problems.

With regards to the major comments:

1. "*The authors need to be clearer regarding what causes thermal offsets. They mention vague processes like 'seasonal variability in thermal conductivity' (P2,L15-20; Section 3.2, Section 3.6.2) but they can be much more explicit. It is not complicated. ...*"
Thank you for this suggestion. We have reformulated the last Paragraph in Section 3.6.2 as:
"Theoretically, thermal offset (TTOP - MAGST) origins from different heat transfer efficiency of an active layer between summer and winter in a permafrost region at equilibrium state. Since the thermal conductivity of ice is four times of water's, it can significantly enhance thermal conductivity of saturated soils from summer to winter. Given a constant and high soil water content, a negative thermal offset will occur due to a much higher thermal conductivity in winter than that in summer. However, the thermal offset can also be negative when the mean total soil water content is higher in summer than in winter. For instance, the positive thermal offset at equilibrium state in Fig. 10 was mainly led by the high ratio of $\lambda_t/\lambda_f$ around 1 via seasonal water content reduction. The schematic mean annual ground temperature profile is closer to the one described by Brown (1970) other than the one suggested by Smith and Riseborough (2002) as shown in Fig. 11. In addition, the concept of thermal offset will be invalid at disequilibrium conditions. For instance, the thermal offsets in Fig. 10 decrease dramatically along with climate warming. It is obvious that the decreasing negative thermal offsets were not caused by $\lambda_f > \lambda_t$ but by the lag between surface and subsurface warming. This is corroborated by the observed thermal offsets (Fig. 4) that positive values occurred in 2007 and 2008 and they decreased to negative in 2009 and 2013. Therefore, the concept of the normal offset is not suitable for the studied case, and the plausible "normal" thermal offset might not necessarily be attributed to $\lambda_f > \lambda_t$, but to permafrost disequilibrium."

2. "*The authors jump right into permafrost thaw in the introduction without any mention of why it is important. I understand that this is a cold regions journal, but some context would be nice. I think the authors should highlight that permafrost thaw has considerable implications for surface and subsurface hydrologic routing (e.g. Kurylyk et al., 2014 ESR), geotechnical failures (Harris et al., 2009, already cited), and carbon dioxide and methane release (Schuur et al., 2015). Those papers are broad review papers on permafrost thaw impacts.*"

Introduction of permafrost thaw impacts are added in the Introduction section.

3. "*Section 3.1. I don't understand how there could be a 5℃ difference between the mean air temperature and mean surface (or very close to surface) temperature. Such differences are commonly observed (e.g. Zhang et al., 2005) but only in regions that experience deep snowpack (Zhang, 2005). This seems like a very high difference without considerable snowpack. Could the authors explain?*"

The notable difference between the mean air temperature (MAAT) and mean ground surface temperature (MAGST) should be attributed to the low absolute humidity in the air at altitude above 4400 m and diurnal freeze-thaw process in the near-surface soils at the study site. As pointed out in the Introduction section, a strong diurnal forcing with some 180 freeze-thaw days (Yang et al., 2007) is very typical on the Qinghai-Tibet Plateau (QTP). Generally, the low absolute humidity can lead to a large diurnal dynamic range in air temperature. As a result, the lower humidity in winter leads to a

bigger diurnal range of air temperature, e.g., > 20℃ than that in summer at the study site. In addition, near-surface soil freeze-thaw process strengths the difference. Particularly, the difference in winter is much bigger than that in summer, because the phase change in the frozen soils takes a large amount energy.

Figure 3.5 shows an example of the daily differences between MAAT and MAGST, and their diurnal dynamic ranges at the study site. P1 and P2 shows two monitoring locations. More details can be found in Pan (2011).

Reference: Pan X.: Hydraulic and Thermal Dynamics at Various Permafrost Sites on the Qinghai-Tibet Plateau. PhD thesis: P57. www.ub.uni-heidelberg.de/archiv/11934

4. "*Section 2.4.1. Where does the water go in the model if no lateral flows are allowed (P4, L11) and no vertical drainage is allowed out of the bottom (P4, L23)? Are ET and P presumed to be balanced? This is very confusing. Also, P4, L26 implies that the thermal conductivity and porosity are used to compute the geothermal flux, when it is really the conductivity times the gradient. Also, in this paragraph and in many other cases through the manuscript, the authors use the term 'soil matrix' to refer to the solid particles. Sometimes matrix (in the context of thermal conductivity) means the matrix of water and solid grains. I recommend that here and elsewhere the authors change the terminology from soil matrix to 'soil particles' or 'soil grains'.*"

In the 1D GEOtop model, water balance of the soil profile is mainly controlled by the ET, P and soil water storage. Since potential evaporation is usually much bigger than the actual evaporation on the QTP, overland flow is rare for the studied case.

The calculation of the geothermal flux was correct ($1.95*0.07 \approx 0.14$), where we calculated it as frozen soil as

$$\lambda_b = \left[(1-0.2)\times\sqrt{2} + 0.05\times\sqrt{0.567} + 0.15\times\sqrt{2.29}\right]^2 = 1.95$$

However, since the measured gradient was derived from the measurements below permafrost base, we should calculate it for the thawed soil. Thus, $\lambda_t = 1.64$ and the geothermal flux is 0.11 W m$^{-2}$. It is corrected in all simulations in the revised manuscript. Thank you for pointing out this issue.

The soil matrix is replaced with soil particles in the revised manuscript.

5. "*The modeling results (simulated vs. observed, Fig. 8) are not good. This modeling exercise certainly did not 'validate' (P8, L2) their model. Observed warming is about twice the modeled rate. The authors propose that this is due to (1) not accounting for snow, (2) difference in simulated and observed climate data, and (3) model assumptions. Okay I can buy that. But why not examine (1) by comparing measured and simulated surface temperatures? In other words, if the problem with the model occurs at the atmosphere-soil interface, then you can easily demonstrate that by comparing the measured and simulated temperature at this point. Also, why did the authors use climate model data for a period when they had site data? This makes no sense at all. Just use the climate data for the future period, not for the model performance assessment period. So I believe the authors can easily test (1) and (2) above.*"

We apologize for misusing the word "validate".

Regard to the suggestion (1), the appended Figure 1 shows the measured and simulated near-surface temperature. Evident deviations occur in the winter, which lead to lower surface temperatures over the range of -7 ~ 5℃. This should be caused by occasional snowcovers.

Regard to the suggestion (2), we partially agree with that. Firstly, to make the simulation more realistic, the climate model data were also adapted with local observations (Figure 7). Therefore, the forcing data may provide an opportunity to mimic the observed permafrost warming for the observed period. Secondly, the aim of comparing the observed and simulated warming rates over a 9-year period is to assess the model performance for the whole period from 1980 to the observed period. Although the results are not so good, it exposes the suggested limitations of our simulations. Thirdly,

Figure 8 also provides a hint for the small differences in the thermal regimes for the three soil architectures in Figure 12.

6. "*Section 3.6.3 and Figure 12, The differences in the thermal regimes for the three soil architectures (A1, A2, and A3) seems rather minor in my opinion (for both the left and right columns in Figure 12). Since this is a major point of the paper, I'm left wondering, 'what's the point'? The series are virtually indistinguishable from 2040 onwards.*"

We agree with your comment that they are no so significant in Figure 12. The reasons are given as follows.

The effect of the seasonal soil moisture reduction on $\lambda_t/\lambda_f$ is mainly controlled by the thermal conductivity of soil particles and the seasonal soil moisture reduction. Since $\lambda_t/\lambda_f$ is very sensitive to the seasonal reduction amount and initial water content, the effect of the seasonal soil moisture reduction on $\lambda_t/\lambda_f$ is related to the soil architecture. The appended Figure 2 gives an example of the impact of summer water content on $\lambda_t/\lambda_f$ . Given a constant moisture reduction of 10%, the smaller the summer water content, the higher the $\lambda_t/\lambda_f$. Unfortunately, given current parameterization it is challenging for our model to capture the field reality, as a result, the simulated seasonal soil moisture reduction is not as significant as the observed one (Figure 9c v.s. Figure 9c'). In Figure 12, the differences are bigger in the right column than in the left column, and this is because of the smaller thermal conductivity of soil particles. While A3 is warmer than A1 and A2, particularly in the right column. The soil architecture A3 does form a dry middle layer (around 1 m in Figure 9c), where smaller seasonal soil moisture reduction could lead to a higher $\lambda_t/\lambda_f$.

In general, Figure 12 demonstrates the effect of soil architectures with the different seasonal soil moisture reductions on permafrost warming, although it is not perfect. Besides, it is possible to get more significant differencesif the simulated seasonal soil moisture reduction is more accurate by using better parameterization for hydraulic properties in future.

The results from 2040 onwards provide a hint of the persistence of our hypothesis over time (see reply to the specific comment 7 in #RC 1), apart from the point that the stratified active layers and associated seasonal soil moisture reduction enhance the high-altitude permafrost warming.

Minor comments

1. *Title, delete 'the' as it is not needed and sounds funny*
   Done

2. *P1, L16-17, delete 'with thickness larger than some 1.5 m' and insert '(>1.5 m) after 'thick'*
   Done

3. *P1, L17, 'It is additionally furthered' is unclear. This should be something like 'The conductivity ratio can be further increased' or something like this*
   Done.

4. *P1, L23, 'high mountains' should be 'alpine regions'. The Alps is not a mountain, for example.*
   Done.

5. *P1, L26, delete 'the' after 'Generally' and 'the' after 'than'*
   Done.

6. *P1, L28, 'Their response': : :.whose response?*
   It is rephrased as "Responses of permafrost controlled by these local factors to climate change is thus also expected to differ."

*7. The last sentence on P1 should be moved before the sentence beginning with 'For instance' The first paragraph reads a bit like a Wu et al. fan club press release. I think it would be better to incorporate some of the implications of permafrost thaw in this paragraph (see major comment above)*
    Rephrased in the revised manuscript.

*8. P2, L3, The reported permafrost warming rate is half of the air temperature warming rate (we would expect it to be lower, so that is fine); however, the paragraph reads as if the permafrost warming rate is higher. I'm not convinced that it makes sense to compare the permafrost warming to air temperature warming over a 13 year period (2002-2014). The subsurface warming rate is lagged (and typically damped) in response to a surface (or atmospheric) warming period. The lag is not that important when you are talking about a 100 year period, but it certainly is over 13 year period.*
    We agree with that it is problematic to compare the warming rates of permafrost and air temperature without considering time lag. Nevertheless, that is cited from the referenced paper. But the inference should be reasonable if the indicated the background of climate warming for the past few decades on the QTP is added. The sentence is rephrased as " Comparing to the rate of 0.3°C per decade for the past five decades over the QTP (Piao et al., 2010), there was no extraordinary increase in air temperature ($0.02°C\ yr^{-1}$) at the investigated area. Thus, an average increasing rate of $0.01°C\ yr^{-1}$ at 10-m depth in permafrost temperature sounds rather high.".

Piao, S.L., Ciais, P., Huang, Y., Shen, Z.H., Peng, S.S., Li, J.S., Zhou, L.P., Liu, H.Y., Ma, Y.C., Ding, Y.H., Friedlingstein, P., Liu, C.Z., Tan, K., Yu, Y.Q., Zhang, T.Y., and Fang, J.Y.: The impacts of climate change on water resources and agriculture in China, Nature, 467, 43–51, 2010.

*9. P2, L8, insert 'the' before 'atmosphere'*
    Done

*10. P2, L11, I'm confused by the comment regarding diurnal forcing and freeze-thaw days. Permafrost is not really diurnally forced.*
    See responses for major comment 3. The diurnal freeze-thawing modifies the coupling between atmosphere and land surface, as well as permafrost.

*11. P2, somewhere the authors could consider citing Hayashi et al. 2007 who proposed a Stefan type algorithm to deal with the problem the paper focuses on (i.e. Changing moisture content through the season).*
    *Hayashi et al. 2007. A simple heat conduction method for simulating the frost-table depth in hydrological models. Hydrol. Process. 21(19)*
    Done

*12. P2, L22, Delete 'the' before 'permafrost'*
    Done

*13. P2, L23 Delete 'a' before 'recent'*
    Done

*14. P2, L25, I don't think diagnose is the right word here: : :.maybe characterize?*
    Done

*15. P3, L12. If the permafrost is 25 m, than the soil at a depth of 10 m must be permafrost. So of course, the temperature would have to be less than 1.0C. In fact, it would have to be less than 0C or it is not permafrost.*
    Missing a sign "-". Corrected.

16. *Heading for section 2.2 contains an extra 'Subsection'*
    Done.

17. *P3, L27, Delete 'the' before 'GEOtop'. Also this sentence would read much better if 'surface and soil, and the soil freezing' were replaced with 'surface and soil as well as the soil freezing and thawing'. Otherwise it sounds like freezing and thawing is included in the list containing atmosphere, surface, and soil.*
    Done.

18. *P3, L29, Change 'allows to simulate' to 'simulates'*
    Done

19. *P3, L30, Delete 'the' before 'complex'*
    Done

20. *P4, L2, the concept of relating the soil freezing curve and soil drying curve is quite foreign to most permafrost scientists. Consider citing the review on this topic.*
    *Kurylyk and Watanabe. 2013. The mathematical representation of freezing and thawing processes in variably-saturated, non-deformable soils. Adv. Wat. Res. 60, 160-177*
    Done

21. *P4, L2, change 'allows' to 'enables the user'*
    Done

22. *P4, L11 delete 'given as'*
    Done

23. *P4, L15, change 'with a high resolution in size of 10 cm: : :and was gradually reduced' to something like 'with elements with a height of 10 cm: : :and reducing to: : :' or something like that*
    Done

24. *P5, L14-16, This is a fragment and confusing*
    It is moved and merged into the introduction section.

25. *P5, L18, insert 'the' before 'active layer'. Insert 'the' after with*
    Done

26. *P5, L24, change 'on' to 'of the'*
    Done

27. *P5, L27. There should be an appropriate citation for CMIP5. If I remember correctly, there is a brief paper published describing the dataset*
    Done.

28. *P5, L32, change 'a quick' to 'the rapid'*
    Done

29. *P6, L3. I'm curious how many 10 year periods were run for the spin up (i.e. how many cycles). This should be mentioned.*
    We used 15 times of 10-year period for spin up. It is added in the revised manuscript.

30. *P6, L11, delete 'well-fitted'*

Done

31. *P6, L11, insert 'in the very shallow subsurface' after 'heat transfer'*
Done

32. *P6, L16, insert 'mean annual' before 'thermal profiles. Delete '. It covers the mean annual temperature data', i.e. combine first two sentences into 1.*
Done

33. *P6, L21. MAGT should be MAGST shouldn't it?*
*Done*

34. *P6, L26, insert 'with the fact' after conflict*
Done

35. *P6, L27, change 'exists' to 'exhibits'*
Done

36. *P7, L11, change 'else' to 'otherwise'*
Done

37. *P7, L12, change 'from' to 'that'*
Done

38. *Last sentence in P7 sounds like it should be in introduction not 3/4 of the way through the paper*
*R*emoved in the revised manuscript.

38. *P8, L7, Delete 'it is higher: : :..Wu et al. (2015)'. This is not relevant given how different the periods are.*
Done

39. *P8, L31, 'validated' should be 'investigated' or something like this*
Done

40. *P9, L7, change 'contrast' to 'the'*
Done

41. *P9, L13, 'underestimated permafrost temperature' is not really a good physical explanation for why thawing is slower in the model than in observations. Of course, this is caused by underestimated permafrost temperature, but the question that should be addressed is 'why is the permafrost temperature underestimated?'*
Revised in the manuscript as "This is mainly related to the different soil water content distributions between observed and simulated ones. The overestimated soil water storage in the shallow layer in the simulation lagged its thawing rate. "

42. *P9, L18, I'm confused by the statement 'and disappears till talik present'*
Revised as "and disappears when talik presents".

43. *P9, L26, 'more close' should be 'closer'*
Done

44. *P10, L8, delete 'to'*
Done

*45. P10, L17-19, this is a fragment*
 It is removed in the revised manuscript.

*46. P10, L20, is this 'extraordinary permafrost warming rate' referring to observed or simulated warming?*
 That's observed one. Revised in the revised manuscript.

*47. P10, L25-27. This sentence seems to contradict itself (although I know what the authors mean): 'In contrast to the normal offset caused by the seasonally variable thermal conductivity, a reversed thermal offset at equilibrium state is formed due to the remarkable high ratio of seasonal thermal conductivity*
 It is rephrased in the revised manuscript as
"In contrast to the normal thermal offset caused by the low ratio of seasonal thermal conductivity, a reversed thermal offset at equilibrium state is formed due to the remarkable high ratio of seasonal thermal conductivity, namely close 1.0 or even higher, given such a weather pattern and soil properties."

*48. Figure 4 – different colours for the series would be helpful (after all TC is all online anyway)*
 Done

*49. Figure 6 caption. The thermal conductivities in (b) are calculated via Eq. (1) right? If so, this should be stated in the caption. Also, how is the ice content obtained for this equation? Somewhere it is stated that the moisture content is assumed to stay the same in the winter. So then the ice is calculated as the total minus liquid?*
 Yes, The thermal conductivities were calculated with Eq. (1). Added in the caption in revised manuscript. The ice content is calculated as soil water content measured just before freezing minus the measured liquid water content in frozen soils. Here we assume the soil moisture migration is negligible due to the coarse soils.

*50. Figure 7 caption: change 'on' to 'of' in both places.*
 Done

*51. Figure 9, 10, and 12. The authors should clearly highlight the differences between the left and right columns (Figure 9 and 12) and the top and bottom (Figure 10). I think it is better to label the figure panels rather than put this info in the caption. Otherwise the reader is scanning up and down.*
 Done.

*52. Table 1. How was the solid particle thermal conductivity of 5 W/(m K) chosen? This is rather high for sand grains in my experience.*
 The value was estimated from local time series of soil temperature measurements. The apparent thermal diffusivity can be inversely estimated by using transfer function, then the effective thermal conductivity can be calculated with thermal properties of soil components. More details can be found in Pan (2011). Finally, the thermal conductivity of the soil particle can be inversely estimated with Eq. (1).
 Reference: Pan X.: Hydraulic and Thermal Dynamics at Various Permafrost Sites on the Qinghai-Tibet Plateau. PhD thesis: P57. www.ub.uni-heidelberg.de/**arc**hiv/11934

[Figure]

**Figure 3.5.** Characteristics of the air temperature $T_a$ (2.0 m above the ground surface) and the ground surface temperature $T_s$ (0.05 m below the surface) at Chumaer. (a) the measured air temperature and the ground surface temperature; (b) the differences between air temperature and ground surface temperature; (c) the diurnal range of air temperature and ground surface temperature.

[Figure]

**Append Figure 1** Comparison of the simulated near-surface soil temperature ($T'_{5cm}$) and observed one ($T_{5cm}$) over the period of 2006-2014.

[Figure]

**Append Figure 2** Influence of the summer soil moisture content and thermal conductivity of soil particles ($K_{sp}$) on the ratio of seasonal thermal conductivity ($\lambda_t/\lambda_f$).

---

## Author Comment (AC2) · 18 Mar 2016

We thank the reviewer for her/his insight comments and suggestions, which significantly enhance the quality of our paper. Regard to the suggested quantitative arguments and the implications of the findings, we have added them in the revised manuscript.

With regards to the major comments:

1. "*The formula for « thermal offset » and « surface offset » should be recall (in the introduction) for better clarity. In section 3.2, confusion is introduced about « thermal offset » : it was defined in the introduction as « TTOP – MAGST ». In section 3.2 it is approximated by « T(-2.18 m below surface) – MAGT » with MAGT quite different from MAGST. Please clarify.*"

As suggested, the thermal offset and surface offset are recalled in the revised manuscript. The "MAGST" was accidentally written as "MAGT" in that sentence. It was replaced with "MAGST" in the revised manuscript.

2. "*Section 2.2 is entitled : 2.2 Surface-subsurface monitoring scheme Subsection. « Scheme subsection » could be deleted from the title.*"
The "subsection" is removed.

3. "*The defined soil architectures in Section « 2.4.2 Simulation protocol » are not consistent with the caption of Fig 9 and the explanations of Section 3.6.1. Please make sure the Architecture definition is consistent in the whole document (maybe add a Table).*"
Thank you very much for pointing out this error. It is corrected in the revised manuscript and the Table 2 is also reformulated as appended.

4. "*P8 l 2, L31 : neither the model nor the effect are 'validated' in the current state of the paper. The comments below may give some sense to the validation of the effect through modelling.*"
We apologize for misusing the word "validate". It is replaced with "characterized".

5. "*Concerning the local $\lambda t / \lambda f$ ratio : Year 2008 is used as an illustration of typical annual conditions. Given that ground temperature and soil water content are being measured at this site since 2006, stepping back from Year 2008 and bringing an interannual perspective would strenghen the paper's conclusion. I at least recommend a Table with the maximum $\lambda t / \lambda f$ value over the upper 2.18 m of the soil for each year with observations.*"
Appended table 3 is added, and two sentences are inserted as "The inter-annual variation of the maximum $\lambda_t/\lambda_f$ in the profile is listed in Table 3. In most years, it is very close or over 1, e.g., 2008 and 2009. The smaller value 0.90 in 2013 is attributed to the wet year with extraordinary rainfall."

6. "*Concerning the impact of the $\lambda t / \lambda f$ ratio on permafrost warming :*
   *o Fig 6 could provide the vertical profiles for $\lambda t$ and $\lambda f$ with $\lambda m=2.5$ W/m/K, in support of the assessment : «In order to exceed the ratio of 1, the seasonal liquid water content has to fall below a certain threshold, which depends on soil thermal conductivity and water content in thawed state. For instance, the soils with high thermal conductivity of soil matrix will need larger liquid water content reduction than that of the soils with small thermal conductivity of soil matrix.»*"

The $\lambda_t$ and $\lambda_f$ in Figure 6 were calculated with $\lambda_m$=5.0 W/m/K. It shows the maximum ratio $\lambda_t/\lambda_f$ >1 at the depth of around 1 m. In most years, it is very close or over 1 (Table 3).

In our simulations, $\lambda_t/\lambda_f$ are relative lower than the observed ones. Given current parameterizations, it is difficult to reproduce comparable hydraulic and thermal dynamics as observed in the active layer. Particularly, the used hydraulic parameters in our model are just determined from some statistical relationships, which are rather rough. Therefore, our simulations can only capture the seasonal pattern of soil moisture reduction, but the absolute amount of seasonal soil moisture amount, as well as water content in thawed state, are still not good enough. Since $\lambda_t/\lambda_f$ is very sensitive to the seasonal reduction amount and soil moisture content in summer, the simulated $\lambda_t/\lambda_f$ is relative low. That's also the major reason for the small differences of permafrost regimes among A1, A2 and A3 (Figure 12). Concerning the relative low impact of soil moisture reduction on $\lambda_t/\lambda_f$ for the simulations using $\lambda_m$=5.0 W/m/K, simulations using $\lambda_m$=2.5 W/m/K are better for visualizing the effects of stratified active layer on permafrost warming.

o *"A high $\lambda_t/\lambda_f$ ratio is advanced as an important argument for an enhanced permafrost warming rate at the observation site. However, Fig. 12 is the only illustration supporting this thesis (as modelling - Fig 8 - fails to reproduce the observed warming) ; it shows that permafrost warming rate is enhanced in the A3 configuration ; the authors explain that this is due to higher $\lambda_t/\lambda_f$ ratio, but this ratio is unfortunately never explicited. I highly recommand adding the mean interannual $\lambda_t/\lambda_f$ for each of the 10-year periods preceeding the selected years of Fig. 12, and for each soil architecture. This would make the paper's main argument less vague. This point is a Major Comment."*

The decadal mean $\lambda_t/\lambda_f$ are added in append Figure 13. The simulations in Fig. 8 fail to reproduce the observed warming because of the passable representation of the land-atmosphere coupling in winter (see more details in reply for #RC 1.) as well as limited representation of hydraulic and thermal dynamics in the active layer. However, simulations in Figure 12 support our hypothesis that the stratified active layer with seasonal soil moisture reduction enhances permafrost warming, although they are not so evident. The added Figure 13 further consolidates it. Overall, the active layers with $\lambda_m$=5.0 W/m/K (left column) have higher values of $\lambda_t/\lambda_f$ than that with $\lambda_m$=2.5 W/m/K (right column), they also lead higher warming rates (Figure 12). However, the smaller differences among the architectures in the left column indicate that the impacts of the seasonal soil moisture reduction on $\lambda_t/\lambda_f$ are not as significant as that in the right column. Generally, the A3 have higher values of $\lambda_t/\lambda_f$ than the others in both columns at the early stage, and they also have relative higher warming rates.

o *"P 10 l 13 : the formulation could be improved (like : high -> higher)"*
Done.

7. *"P5 l 30 and P8 l 24 : a crucial thing is to know whether the annual cycle of precipitation in the chosen downscaled projections, is still monsoon-like (as today) or shifts to different patterns in future climate. The authors mention that the projected rainfall may not be accurate. However, given the importance of the annual rainfall pattern on the site specific sub-surface thermal dynamics, **more investigations** on the projected precipitation pattern in the chosen downscaled climate product is needed, in support of the assessment of the impact of $\lambda_t/\lambda_f$ on the warming. This point is a Major Comment."*

We agree with that the monsoon-like precipitation pattern is crucial for the site specific sub-surface thermal dynamics. The prevailing view for the Asian summer monsoon is caused by the elevated heat source driven by the QTP (e.g., Yeh et al., 1957, Yanai et al., 1992). Observations of energy fluxes over the QTP show climate change may led to a shift of monsoon intensity (Duan et al., 2011). However, the Asian monsoon mechanism might not disappear in the near future due to the dominant factor of high elevated topography. Although the monsoon precipitation intensity might shit

a little bit, the annual rainfall pattern will still lead to a notable seasonal soil moisture reduction, as well as high ratio of $\lambda_t/\lambda_f$ in the active layer.

In addition, as explained for the general comments this study is aim to address the effects of stratified active layers on the high-altitude permafrost warming, the monsoon precipitation pattern is a precondition. The selected climate model does reproduce a monsoon-like precipitation pattern for the whole period (1850-2100). It assumed that the monsoon was not changed in the future. Since we are not aim to predict the permafrost warming but to test our hypothesis that the stratified active layer enhances permafrost warming due to the high ratio of $\lambda_t/\lambda_f$, the selected climate model data is reasonable.

To make clarify the persistence over time of the hypothesis, we add a paragraph to clarify this issue in the revised manuscript as follows.

" In addition, the effects of stratified active layers on permafrost warming also evolve in time. As a key role in enhancing $\lambda_t/\lambda_f$, the seasonal soil moisture reduction can be influenced by changes in precipitation pattern and active layer thickness. The Asian summer monsoon, caused by the elevated heat source driven by the QTP (e.g., Yeh et al., 1957, Yanai et al., 1992) may not disappear in the near future, but the monsoon intensity might shift due to climate change (Duan et al., 2011). In addition, change in active layer thickness will influence the suprapermafrost water level, which is essential to the soil water content distribution in the active layer. Given a monsoon dominated precipitation pattern in the projected climate model data, differences in the evolution of $\lambda_t/\lambda_f$ in the shallow layers (0-1.5m) are shown in Fig. 13. Compared to the same soil architectures but $\lambda_{sp} = 2.5$ W m$^{-1}$ K$^{-1}$, the decadal mean $\lambda_t/\lambda_f$ in the left column are higher than that corresponding in the right column. But the differences of $\lambda_t/\lambda_f$ amount the architectures are bigger in the right column. Besides, the highest values of $\lambda_t/\lambda_f$ in A3 are bigger than the other two from 1980s to 2040s, then they gradually become smaller. Generally, the shrinking differences in decadal mean $\lambda_t/\lambda_f$ among the three architectures indicate that the effects of the stratified active layers on permafrost warming are significant in the early state of permafrost degradation, and they will decrease afterward. This is one major reason for the small differences in thermal regimes of the three architectures in Fig. 12."

Yeh, T.C., Luo, S.W., and Chu, P.C.: The wind structure and heat balance in the lower troposphere over Tibetan Plateau and its surrounding, Acta Meteor. Sin., 28, 108–121, 1957.

Yanai, M., Li, C., and Song, Z.: Seasonal heating of the Tibetan Plateau and its effects on the evolution of the Asian summer monsoon. J. Meteor. Soc. Japan, 70, 319–351, 1992.

Duan, A.,Li, F.,Wang, M.,and Wu, G.: Persistent weakening trend in the spring sensible heat source over the Tibetan Plateau and its impact on the Asian summer monsoon, J. Climate, 24, 5671–5682, 2011.

**Technical Corrections**

1. "- *Very frequently the authors confuse « whereas » with « while » or « in the opposite ». (p4 l 23 ; p5 l 14 ; p6 l 13 and l 24 ; p9 l 28 ; ...)*"
   Done.

2. "- *P2 l 14 : basing -> based*"
   Done.

3. "- *P3 l 15 : humility -> humidity*"
   Done.

4. "*P4 l 2 : incomplete sentence*"
   It is rephrased as "It invokes a relation between the soil freezing characteristic and the soil water characteristic and assumes a rigid soil scheme without change in volume for water phase transition (Kurylyk and Watanabe, 2013)".

5. "*P5 line 13 to 16 : unclear, please reformulate*"
   It is moved and merged into the introduction section.

6. "*- P9 l 18 : till talik -> when talik*"
   Done.

**Table 2.** Six simulations with different combinations of soil architecture and thermal conductivity of soil particles ($\lambda_{sp}$) for the shallow layer (0-3.0 m). A1, A2 and A3 stand for three types of soil architecture for the shallow layer. I and II stand for the two types of soil properties in Table 1.

| $\lambda_{sp}$ / W m$^{-1}$ K$^{-1}$ | Architecture | | |
|---|---|---|---|
| | A1: I | A2: II | A3: I + II |
| 5.0 | 1 | 2 | 3 |
| 2.5 | 4 | 5 | 6 |

**Table 3.** Inter-annual variation of the maximum seasonal thermal conductivity ratio ($\lambda_t/\lambda_f$) in the monitoring profile. * Hydrological year: from May 1 to the next April 30.

| Year* | 2007 | 2008 | 2009 | 2010 | 2011 | 2012 | 2013 |
|---|---|---|---|---|---|---|---|
| $\lambda_t/\lambda_f$ | 0.99 | 1.01 | 1.01 | 0.97 | - | 1.00 | 0.90 |

[Figure]

Append Figure 13. Comparison of the influence of soil architecture and thermal conductivity of soil particles ($\lambda_{sp}$) on the seasonal thermal conductivity ratio $\lambda_t/\lambda_f$ in the shallow active layer over the period from 1980 to 2100. (a) - (e) Decadal mean $\lambda_t/\lambda_f$ of A1, A2 and A3 with a high $\lambda_{sp} = 5.0$ W m$^{-1}$ K$^{-1}$ at selected decades; (f) - (j) the same as (a) - (e) but with a low $\lambda_{sp} = 2.5$ W m$^{-1}$ K$^{-1}$.

---

## Referee Comment (RC3) · S. Endrizzi (Referee) · 4 Apr 2016

As a developer of GEOtop model I am very happy to review this paper, which applies the model for a permafrost site in the Qinghai-Tibet plateau and shows that the thermal conductivity of the thawed soil in a permafrost site can be in some conditions larger than the thermal conductivity of frozen soil. The topic is very interesting, the model really fits well with it, and the research questions are well posed. My attention was particularly drawn by the model settings and results:

1. The model settings are extremely important since they strongly affect the results. However, the paper does not fully describe them. For example, the paper should list the van Genuchten parameters, since the behaviour during freezing/thawing is based

on them. It is not enough to refer to neural network routines.

2. The characteristics of the 3 soil architectures A1, A2, A3 are not completely clear to me. You should put a table or drawing that clarifies the soil layers with correspondent properties and parameters.

3. In the papers the parameters are assigned in a deterministic way. Apart considering 3 soil architectures, no or little sensitivity to parameters is performed. This is extremely important, since many parameters are actually idealised or strongly simplified. The van Genuchten parameters result from a strongly simplified model of soil retention, and, since the results are dependent on them, a sensitivity analysis is essential. Pedo-transfer functions and, probably, neural network routines have limitations and cannot be fully trusted. The sensitivity to other parameters should also be considered, for example, when no data are available, for bottom soil, snow precipitation, lateral flow, albedo, etc. In addition, you set the vegetation coverage to 0.3, referring to Gubler at al. (2013), but in this paper we did not consider vegetation.

4. The simulation settings also assume simplified conditions that are described only at the end of paragraph 3.5, namely to justify disagreements between observations and model results. The simplifications should be listed at the beginning, and their plausibility discussed in advance.

5. In par. 3.6 you write that the effect of stratified active layer is validated with modelling. Validate is a strong word. You are not validating, but you are using the model to understand physical processes.

6. The formula of Cosenza et al. (2003) is just one parameterisation of bulk thermal conductivity. GEOtop gives also the possibility to use other formulae (De Vries for example). Maybe it would be worth checking if there are significant differences in the results if other formulae are used.

7. In Fig. 6a you consider only unfrozen water content. However, bulk thermal conductivity also depends on ice content. You should discuss this point.

8. I do not understand why in Fig. 5b the 0 C isotherm is not close to the curve of the unfrozen water content decrease.

9. In the paper you often use temperature/time as a proxy of permafrost warming. However, temperature only describes the effect of sensible heat, but not the latent heat. If permafrost has a temperature close to 0 C, more heat is needed to increase soil temperature, because some energy is needed for thawing. Therefore, I do not think that a temperature difference of 0.01 C to end spinup is good. Performing a good spinup is also essential to have good model results. This should be more completely described. For how many years the spinup simulation was run? You should also check that water and ice content differences are small to end spinup.

10. In 2014 I wrote a paper describing the model, in particular the version 2.00. Although you used a previous version, you should have a look and cite the paper. This is the link to the paper:

http://www.geosci-model-dev.net/7/2831/2014/gmd-7-2831-2014.html

11. I saw some errors in the English language. Please correct them.

---

## Author Comment (AC3) · 14 Jun 2016

We thank Dr. Endrizzi for his insight comments of the modelling issue. Regard to the suggestions about model sensitivity analysis, although we could not take all the suggestions in this paper, they will motivate us to do a thorough investigation in future.

With regards to the major comments:

  1. "*The model settings are extremely important since they strongly affect the results. However, the paper does not fully describe them. For example, the paper should list the van Genuchten parameters, since the behaviour during freezing/thawing is based on them. It is not enough to refer to neural network routines.*"

   The van Genuchten parameters are listed in revised Table 1.

2. "*The characteristics of the 3 soil architectures A1, A2, A3 are not completely clear to me. You should put a table or drawing that clarifies the soil layers with correspondent properties and parameters*"
   Soil properties of the three architectures were detailed in Line 12 - 27 in page 5 as well as shown in the revised Table 1 (attached at the end).

3. "*In the papers the parameters are assigned in a deterministic way. Apart considering 3 soil architectures, no or little sensitivity to parameters is performed. This is extremely important, since many parameters are actually idealised or strongly simplified. The van Genuchten parameters result from a strongly simplified model of soil retention, and, since the results are dependent on them, a sensitivity analysis is essential. Pedotransfer functions and, probably, neural network routines have limitations and cannot be fully trusted. The sensitivity to other parameters should also be considered, for example, when no data are available, for bottom soil, snow precipitation, lateral flow, albedo, etc. In addition, you set the vegetation coverage to 0.3, referring to Gubler at al. (2013), but in this paper we did not consider vegetation.*"
   We agree with the reviewer concerning the nature of the model parameters and the validity of methods to estimate them independently. The approach in this paper is to use best available independent information for the simulation. Besides corroborating the general understanding of the observed processes, this also demonstrates the challenges for quantifying situations where data are not available, which is the vast majority, unfortunately.

   The next step will encompass a site-specific sensitivity analysis of the simulation followed by a proper inversion for the parameters. This will then also provide the statistical basis for better assessing the true uncertainties. That next step is beyond the scope of the current paper, however.

4. "*The simulation settings also assume simplified conditions that are described only at the end of paragraph 3.5, namely to justify disagreements between observations and model results. The simplifications should be listed at the beginning, and their plausibility discussed in advance.*"
   Agree. It is revised in the new manuscript.

5. "*In par. 3.6 you write that the effect of stratified active layer is validated with modelling. Validate is a strong word. You are not validating, but you are using the model to understand physical processes.*"
    Agree. See more explanation in the reply for referee #1.

6. "*The formula of Cosenza et al. (2003) is just one parameterisation of bulk thermal conductivity. GEOtop gives also the possibility to use other formulae (De Vries for example). Maybe it would be worth checking if there are significant differences in the results if other formulae are used.*"
    First of all, we do believe that there might be significant differences in the results if we use improper formulae for the bulk thermal conductivity. Here we choose the formula of Cosenza et al. (2003), because it has been verified with some published data in satisfactory agreement both for saturated rocks and for unsaturated soils.

7. "*In Fig. 6a you consider only unfrozen water content. However, bulk thermal conductivity also depends on ice content. You should discuss this point.*"
    We did consider the ice content for calculating the bulk thermal conductivity. The total water content in the caption means the sum of unfrozen water content and ice content. To avoid misunderstanding, the total liquid water content has been replaced with "total water content" in the text.

8. "*I do not understand why in Fig. 5b the 0 °C isotherm is not close to the curve of the unfrozen water content decrease.*"
    This is quite common in field observations. First of all, the freezing point of soil water will be reduced below 0°C due to soil salinity. Secondly, the soil temperature gradient is so small within the zero-curtain that the 0°C isotherm is not close to the curve of the unfrozen water content decrease.

9. "*In the paper you often use temperature/time as a proxy of permafrost warming. However, temperature only describes the effect of sensible heat, but not the latent heat. If permafrost has a temperature close to 0°C, more heat is needed to increase soil temperature, because some energy is needed for thawing. Therefore, I do not think that a temperature difference of 0.01°C to end spinup is good. Performing a good spinup is also essential to have good model results. This should be more completely described. For how many years the spinup simulation was run? You should also check that water and ice content differences are small to end spinup.*"
    We agree that checking temperature difference in conjunction with water and ice content differences would be more reliable. In this study, to reach a temperature difference of 0.01, the spinup simulation runs for 150 years, and the total mean annual water content difference is less than 0.01. This should be fine.

10. "*In 2014 I wrote a paper describing the model, in particular the version 2.00. Although you used a previous version, you should have a look and cite the paper. This is the link to the paper: http://www.geosci-model-dev.net/7/2831/2014/gmd-7-2831-2014.html*"
    Done.

11. "*I saw some errors in the English language. Please correct them.*"
    Done.

**Table 1.** Soil properties of shallow soils (A: 0-3.0 m) and underlying soils (B: 3.0-30 m) for three soil profiles (A1/B, A2/B and A3/B). $K_s$: saturated hydraulic conductivity; $\alpha$ and $n$: van Genuchten parameters; $\theta_r$ and $\theta_s$: residual and saturated soil water content, respectively; $\lambda_{sp}$: thermal conductivity of soil particles; $C$: thermal capacity.

| Soil architecture | | A1 | A2 | A3 | | B |
|---|---|---|---|---|---|---|
| | | 0-3.0 m | 0-3.0 m | 0-0.3 m | 0.3-3.0 m | 3.0-30 m |
| Soil texture % | sand | 66.3 | 92.2 | 66.3 | 92.2 | - |
| | silt | 12.0 | 3.8 | 12.0 | 3.8 | - |
| | clay | 21.7 | 4.0 | 21.7 | 4.0 | - |
| Hydraulic properties | $K_s$ / m d$^{-1}$ | 0.19 | 4.68 | 0.19 | 4.68 | $2.2\times10^{-3}$ |
| | $\alpha$ / cm$^{-1}$ | 0.03 | 0.03 | 0.03 | 0.03 | 0.01 |
| | $n$ / - | 1.33 | 2.85 | 1.33 | 2.85 | 1.5 |
| | $\theta_r$ / m$^3$ m$^{-3}$ | 0.06 | 0.05 | 0.06 | 0.05 | 0.10 |
| | $\theta_s$ / m$^3$ m$^{-3}$ | 0.38 | 0.38 | 0.38 | 0.38 | 0.2 |
| Thermal properties | $\lambda_{sp}$ / W m$^{-1}$ K$^{-1}$ | | | 5.0 | | 2.0 |
| | $C$ / J m$^{-3}$ K$^{-1}$ | | | $2\times10^6$ | | |

---

## Author Response (AR2)

We appreciate the reviewer again for the help in language problems. Detailed replies are list as follows.

1. "*P1, L14, 'smaller' should be 'lower'*"
    Done.

2. "*P2, L6-7, why is the warming rate given per decade at the first of the sentence and then per year later on? Also, these two presented rates only differ by 0.01 C/yr.*"
    Changed them in the same unit ˚C/yr. The difference is due to contrast averaging areas that the former one is for overall Qinghai-Tibet Plateau, whereas the latter one just represent one much smaller mountain permafrost area.

3. "*P2, L9-11, this is an awkward and confusing sentence*"
   It is rephrased as "Under the warming of the atmosphere, the high warming rate of the permafrost is even similar to that of air temperature rise, $0.02$˚C $yr^{-1}$. Thereby, we expect a dominating role of subsurface processes in the active layer that amplify the climate warming input.".

4. "*P2, L20, I'd delete 'in addition to the soil thermal properties' as it is not needed in this sentence, and the sentence has iffy structure*"
    Done.

5. "*P2, L20-22, as I noted last time, and the authors insufficiently addressed, it is not really appropriate to say that permafrost has diurnal forcing. Permafrost does not freeze and thaw on a daily basis. This sentence is confusing.*"
   We delete this sentence.

6. "*P2, L34, change 'highly demanded' to 'required' and 'facilitate' to 'enable' Throughout the manuscript: I'd say 'rapid permafrost warming' not 'quick permafrost warming'*;
    Done.

7. "*This is just a comment and the authors do not need to change anything, but I remain skeptical that analytical models (at least in theory) cannot address the seasonal moisture content/thermal conductivity and layering issue that the authors address. As reviewed by Walvoord and Kurylyk (2016), Stefan equation modifications have been proposed to address soil layering, changing moisture content, sensible heat affects, and two-directional heat transfer. These algorithms can often be combined, and thus many of the analytical assumptions can be relaxed.*

*Walvoord MA, Kurylyk BL. 2016. Hydrologic impacts of thawing permafrost – A review. Vadose Zone Journal, DOI: 10.2136/vzj2016.01.0010*"
   Thank you for suggestion. We will take it for consideration in our future work.

8. "*P3, L29, change 'Thereby' to 'Thus'*"
    Done.

9. "*P4, L1, delete 'They were' and combine the 2 sentences into 1.*"
    Done.

10. "*P4, L23, delete ‛. They are’ and add a colon after ‛drilling’ (i.e. combine into one sentence)*"
   The sentence is rephrased as "The soil profile domain was generated with element size of 10 cm for the shallow soils (0-3.0 m) and reducing to 0.5 m and 1.0 m for the underlying soils. There are 63 elements in total.".

11. "*P4, L26 move ‛solely’ after ‛to’*"
   Done.

12. "*P8, L8, change ‛is rarely addressed’ to ‛has been rarely addressed’*"
   Done.

13. "*P8, L31, change ‛not well accurate in general’ to ‛not generally accurate’*"
   Done.

14. "*P10, L3, change ‛of water’s’ to ‛that of water’*"
   Done.

15. "*P10, L3-4, this statement is not true. It assumes a porosity of 1 (and also ignores changes in the heat capacity of ice vs. water). It is not needed, so just delete it.*"
   Done.

16. "*P10, L6, insert ‛it is’ after ‛instance’*"
   Done.

17. "*P11, L16, change ‛sparsely’ to ‛sparse’*"
   Done.

18. "*P11, L18, this is not really the right use of the term ‛indispensable’. It should be something like ‛instrumental’*"
   Done.

19. "*P11, L21, for this point I would insert ‛negative’ after ‛normal’ and ‛($\geq$ 1)’ after ‛reversed’. I'd also delete "namely close 1.0 or even higher, given such a weather pattern and soil properties"*"
   Done.

20. "*P11, L24, change ‛facilitating to raise’ to ‛raising’*"
   Done.